



**Inter-comparison of O₃ formation and radical chemistry in the past decade at a suburban site in**
**Hong Kong**
Xufei Liu[1,#], Xiaopu Lyu[1,#], Yu Wang[1], Fei Jiang[2], Hai Guo[1,*]
[1] Air Quality Studies, Department of Civil and Environmental Engineering, The Hong Kong
Polytechnic University, Hong Kong, China
[2] Jiangsu Provincial Key Laboratory of Geographic Information Science and Technology,
International Institute for Earth System Science, Nanjing University, Nanjing, China
*Corresponding author. ceguohai@polyu.edu.hk
# Both authors made equal contribution.
**Abstract**
Hong Kong, as one of the densely populated metropolises in East Asia, has been suffering
from severe photochemical smog in the past decades, though the observed nitrogen oxides
($NO_x$) and total volatile organic compounds (TVOCs) were significantly reduced. This study,
based on the observation data in the autumns of 2007, 2013 and 2016, investigated the
photochemical ozone ($O_3$) formation and radical chemistry during the three sampling periods
in Hong Kong with the aid of a Photochemical Box Model incorporating the Master
Chemical Mechanism (PBM-MCM). Neither the observed $O_3$ nor the simulated locally
produced $O_3$ changed significantly ($p$=0.11 and 0.99, respectively) from 2007 to 2013;
however, both of which decreased ($p<0.05$) from the VOC sampling days in 2013 to those in
2016 at a rate of -5.04±0.05 and -4.35±0.10 ppbv yr$^{-1}$, respectively. The regionally
transported $O_3$ showed an increase (rate = 1.62±0.39 ppbv yr$^{-1}$, $p<0.05$) during 2007-2013,
but slight decrease ($p$=0.09) from 2013 to 2016. The mitigation of autumn $O_3$ pollution in this
region was further confirmed by the continuous monitoring data, which has never been
reported in previous studies. Benefited from the air pollution control measures taken in Hong
Kong, the local $O_3$ production rate decreased remarkably ($p<0.05$) from 2007 to 2016, along



with the lowering of recycling rate of hydroxyl radical (OH). Specifically, VOCs emitted
from the source of liquefied petroleum gas (LPG) usage and gasoline evaporation decreased
in this decade at a rate of $-2.61 \pm 0.03$ ppbv yr$^{-1}$, leading to a reduction of the $O_3$ production
rate from $0.51 \pm 0.11$ ppbv h$^{-1}$ in 2007 to $0.10 \pm 0.02$ ppbv h$^{-1}$ in 2016. In addition, solvent
usage made decreasing contributions to both VOCs (rate = $-2.29 \pm 0.03$ ppbv yr$^{-1}$) and local $O_3$
production rate ($1.22 \pm 0.17$ and $0.14 \pm 0.05$ ppbv h$^{-1}$ in 2007 and 2016, respectively) in the
same period. All the rates reported here were for the VOC sampling days in the three
sampling campaigns. It is noteworthy that meteorological changes also play important roles
in the inter-annual variations of the observed $O_3$ and the simulated $O_3$ production rates.
Evaluations with more data in longer periods are therefore recommended. The analyses on
the decadal changes of the local and regional photochemistry in Hong Kong in this study may
be a reference for combating China's national-wide $O_3$ pollution in near future.
**Keywords:** Ozone formation; Volatile organic compounds; Radical chemistry; Source
apportionment; Control measures
**1 Introduction**
Ground-level ozone ($O_3$) is one of the most representative air pollutants in photochemical
smog, produced through photochemical reactions between volatile organic compounds
(VOCs) and nitrogen oxides ($NO_x$) in presence of sunlight (NRC, 1992; Jacob et al., 1999;
Guo et al., 2017). It is well documented that $O_3$ is harmful to human health (Bell et al., 2004),
crops (Wang et al., 2005) and natural ecosystems (Ashmore, 2005). Through the last 30 years,
extensive efforts have been made by the local and federal governments to alleviate the
tropospheric $O_3$ pollution around the world (NRC, 1992; NARSTO, 2000; Wang et al., 2017a;
Wang et al., 2018a). Effectiveness has gradually shown in some countries/regions, such as



Switzerland, Germany, Ireland and eastern North America (Lefohn et al., 2010; Cui et al.,
2011; Derwent et al., 2013; Parrish et al., 2014; Lin et al., 2017). In contrast, the $O_3$ levels in
many places are still increasing or not decreasing at the expected rates, particularly in East
Asia (Ding et al., 2008; Xu et al., 2008; Parrish et al., 2014; Xue et al., 2014a; Wang et al.,
2017a).
Hong Kong, as one of the densely populated metropolises in East Asia, has been suffering
from severe photochemical smog in the past decades, though the locally-emitted $NO_x$ and
total VOCs (TVOCs) were significantly reduced (Xue et al., 2014a; Ou et al., 2015; Lyu et al.,
2016a; Wang et al., 2017a). On one hand, this indicates the non-linear relationship between
$O_3$ and its precursors. On the other hand, in addition to local $O_3$ formation, the observed $O_3$ in
Hong Kong is also influenced by the regional transport due to the proximity of the highly
industrialized Pearl River Delta (PRD) region. Earlier studies revealed that the local $O_3$
production is typically limited by VOCs in urban and some suburban areas in Hong Kong
(Zhang et al., 2007; Ling et al., 2014; Wang et al., 2017b). Namely, cutting VOCs emissions
will reduce $O_3$ production, while the reduction of $NO_x$ may cause an $O_3$ increment (Cheng et
al., 2010, 2013; Guo et al., 2011; Wang et al., 2017a). Previous studies also documented that
photochemical $O_3$ formation is dependent upon the ratios between TVOCs and $NO_x$ (Sillman,
1999; Guo et al., 2013; Ling et al., 2013), reactivity of VOC species (Zhang et al., 2007; Liu
et al., 2008; Cheng et al., 2010) and the composition of $NO_x$ (*i.e.* relative abundances of $NO_2$
and NO) (Richter et al., 2005; Xu et al., 2008; Wang et al., 2018a). Moreover, located in the
subtropical region, Hong Kong has relatively high temperature and strong solar radiation,
which are favourable for local $O_3$ formation. For regional transport, studies (Wang et al.,
2001; Ding et al., 2004; Wang et al., 2017b) indicated that $O_3$ was generally built up in Hong
Kong under the northerly winds, whereas it was often driven down by the sea breeze from
South China Sea (SCS) and by the southwest monsoon in warm seasons. The contribution of



regional transport to $O_3$ in Hong Kong even reached 70% under the dominance of tropical
cyclone (Huang et al., 2005), a typical synoptic condition conducive to severe $O_3$ pollution in
the Northern Hemisphere (So and Wang, 2003; Huang et al., 2005; Lam et al., 2005). To
improve the air quality in Hong Kong, a series of control measures aiming at restriction of
VOC emissions have been implemented by Hong Kong government since 2007, which
effectively reduced the concentrations of some VOCs, such as propane and $i$-/$n$-butanes
emitted from taxis and public light buses fuelled by liquefied petroleum gas (LPG) (Lyu et al.,
2016b), the aromatics mainly attributable to solvent usage, and the alkenes in association
with diesel exhaust (Lyu et al., 2017a).  As a result, Xue et al. (2014a) and Wang et al.
(2017a) found that the locally produced $O_3$ decreased. However, the regional and super-
regional transport of $O_3$ and its precursors from PRD and eastern China to Hong Kong had
offset the decrease of the local $O_3$ production, resulting in an overall increase of the observed
$O_3$ in Hong Kong from 2005 to 2013.
Despite many previous studies (Xue et al., 2014a, 2016; Ou et al., 2015; Lyu et al., 2016a;
Wang et al., 2017a; Wang et al., 2018a), the inter-annual variations of the $O_3$ formation
regimes and radical chemistry have yet been fully understood in Hong Kong. Additionally,
the online measurement data used in previous long-term $O_3$ study might hamper the exact
understanding of the local $O_3$ formation mechanisms, due to the unavailability of many
reactive VOCs, such as formaldehyde. Besides, the trends of the local production and
regional transport of $O_3$ were only updated to 2013 in previous studies (Xue et al., 2014a;
Wang et al., 2017a). In fact, many measures were taken to reduce air pollutants' emissions in
the latest years in Hong Kong and PRD. For examples, nearly 75% of the old catalytic
converters on LPG-fuelled vehicles were renewed during September 2013 - May 2014. A
program to eliminate the pre-Euro IV diesel vehicles or to upgrade their emission standards to
Euro IV was initiated in March 2014 and is still ongoing till 2019 at its third phase. In PRD,





the second stage of the clean air controlling program was implemented in 2013 - 2015
(DGEPD, 2013). In 2014, the Guangdong provincial government has launched an Action
Plan for Air Pollution Prevention and Control (MEE PRC, 2014), putting the emphases on the
emission control of traffics, coal-fired power plants and industrial sources. Investigations on
the post-2013 variations of the local $O_3$ production in Hong Kong and the regional impacts
provide a good opportunity for us to examine the effectiveness of these local and regional
measures.
The objectives of this study were to re-examine the $O_3$ trend in the pre-2013 and trace the $O_3$
evolution in the post-2013 in Hong Kong, and to explore the underlying mechanisms for the
variations of $O_3$ formation and radical chemistry. With the aid of a photochemical box model,
the locally-produced and regionally-transported $O_3$, as well as their variation trends, were
determined. Under the assumption that the local $O_3$ production in these years was changed
due to a series of control measures in Hong Kong, we also aimed to evaluate the actual
effectiveness of these control measures. China is suffering from severe $O_3$ pollution, almost
second to none over the world. While $O_3$ began to decrease in most areas of North America
and Europe, China's $O_3$ pollution was even aggravated in recent years. A series of air
pollution control strategies have been implemented in China, though most of them were not
specifically designed for $O_3$ abatement. Investigations on $O_3$ trends and the potential causes
in Hong Kong would provide a good example of assessing the evolution of $O_3$ pollution and
the effects of artificial interventions in China. In addition, the changes in the regional
contribution to $O_3$ in Hong Kong determined in this study would throw light upon the
variations of $O_3$ in China, particularly in South China. It is expected that this study would
have some inspiration to the $O_3$ pollution control in other cities and regions in China.
**2 Methodolgy**



## 2.1 Sampling site

Hong Kong is located on the southern coast of China with Guangdong province to the north and Pearl River Estuary (PRE) to the northwest. The sampling site (22.29N, 113.94E), Tung Chung (TC), was in a newly-developed suburban area in western Hong Kong, with a population of ~77,400 in 2016 (CSD, 2011, 2018). The urban centre of Hong Kong is ~20 km northeast of TC. Hong Kong is dominated by the subtropical oceanic monsoon climate. During warm seasons, the prevailing winds mainly come from SCS at a relatively low speed (southwest winds). In cold seasons, the east and northeast winds are predominant. Generally, the sampling site receives relatively polluted air masses from mainland China, *i.e.* PRD region, Yangtze River Delta region and even North China between October and March, when high $O_3$ levels are often observed (Wang et al., 2009). Therefore, the samplings were mainly conducted in October and November in this study, except for 4 out of 45 sampling days in September. The sampling site was close to a highway linking to the Hong Kong International Airport (HKIA), and the HKIA was around 3 km to the north of the site. In addition, the local emissions from residential activities may modulate the air quality at this site. Figure 1 shows the locations of the sampling site (TC) and the 12 air quality monitoring stations in PRD, which witnessed the evolution of air quality in PRD over the last decade and is used to demonstrate the variations of regional $O_3$ in this study. More detailed description of the site can be found in our previous studies (Jiang et al., 2010: Cheng et al., 2010; Ling et al., 2013; Ou et al., 2015).




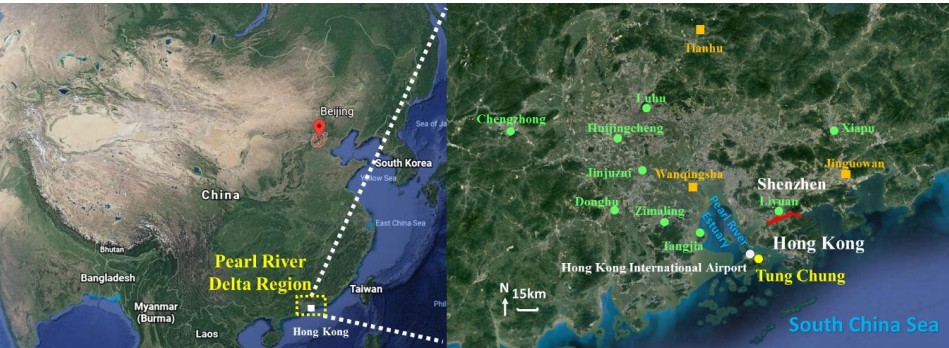


Figure 1. Location of the sampling site (yellow circle) and the surrounding environment. The
red line in the right panel shows the border between Hong Kong and Shenzhen, Guangdong.
The three regional and nine urban air quality monitoring stations in PRD are symbolized by
orange blocks and green circles, respectively.
**2.2 Continuous measurements of trace gases and collection of VOC/OVOC samples**
Trace gases ($SO_2$, CO, NO, $NO_2$ and $O_3$) and meteorological conditions were continuously
measured at TC site for three autumn periods in 2007, 2013 and 2016 (see Table S1 for the
specific sampling periods), including 25 $O_3$ episode days with the maximum hourly average
$O_3$ exceeding 100 ppbv (Level II of China National Ambient Air Quality Standard) and 185
non-episode days. VOC and OVOC samples were selectively collected on 8, 19 and 18 days
in 2007, 2013 and 2016, respectively (see Table S1 for the specific sampling dates). The three
sampling periods were used as representatives of the autumns in the three years in this study,
and the rationality will be discussed in section 3.1.
Trace gases were continuously measured at the TC air quality monitoring station operated by
the Hong Kong Environmental Protection Department (HKEPD), ~0.8 km to our sampling
site. The instruments were the same as those used in the US air quality monitoring program
(HKEPD, 2017a). Table S2 summarizes the instruments, analysis techniques, detection limits
and the time resolutions for measurements of the trace gases. The high resolution data were



collected and averaged into the hourly averages. All the analysers except $O_3$ analyser were
zeroed daily by analysing scrubbed ambient air and calibrated every two weeks by a span gas
mixture with a NIST (National Institue of Standards and Technology) traceable standard,
while the $O_3$ analyser was calibrated using a transfer standard (Thermo Environmental
Instruments (TEI) 49PS) every two weeks. Details about the quality assurance and control
procedures can be found in Ling et al. (2016a). The meteorological parameters, including
temperature, relative humidity, pressure, wind speed, wind direction, precipitation and solar
radiation, were also continuously monitored by a mini weather station (Vantage Pro TM &
Vantage Pro 2 Plus TM Weather Stations, Davis Instruments) during the sampling periods.
Data were integrated into 30-minute averages by a built-in program in the weather station.
The collection and analysis of VOCs and OVOCs were detailed in our previous studies (Guo
et al., 2009; Wang et al., 2018b). Briefly, pre-cleaned and evacuated 2 L electropolished
stainless-steel canisters were used to collect VOC samples. On $O_3$ episode days, one-hour
sample was collected in each hour during the daytime (07:00-19:00 LT), generating 13
samples per day, while 5-7 one-hour samples were collected every other hour on non-$O_3$
episode days from 07:00 to 19:00 LT in the 2013 and 2016 sampling campaigns. However,
12 one-hour samples were collected on each VOC sampling day between 07:00 and 18:00 in
2007, regardless of $O_3$ episodes or non-episodes. The $O_3$ episode days were predicted prior to
sampling based on weather forecast and numerical simulation of $O_3$. Overall, the $O_3$ episodes
were usually associated with high temperature, strong solar radiation, low humidity, and
weak or northerly winds. A total of 414 canister samples, including 96 samples in 2007, 146
samples in 2013 and 172 samples in 2016, were collected and analysed during the three
sampling periods (Table S1).
In addition to VOC samples, OVOC samples were also collected on the same days as those
for the collection of VOCs. Dinitrophenylhydrazine (DNPH)-silica cartridges (Waters Sep-





Pak DNPH-Silica, Milford, MA) were used to collect the OVOC samples. An ozone scrubber
(Sep-Pak; Waters Corporation, Milford, MA) was connected in front of the DNPH cartridge
to prevent interference of ozone. The ozone scrubber was replaced every two OVOC samples.
For each OVOC sample, air was drawn to pass the $O_3$ scrubber and the cartridge for 2 hours
(2.5 hours in 2007 sampling campaign) at a flow rate of 0.5 L $min^{-1}$, which was controlled by
a rotameter. During the sampling periods in 2013 and 2016, 5-7 OVOC samples were
collected every two hours from 06:00-20:00 LT on both $O_3$ episode and non-episode days. In
2007, only 2 samples were collected on non-$O_3$ episode days at 10:30-13:00 and 13:00-15:30,
and 4 samples between 08:00 and 18:00 on $O_3$ episode days. In total, 275 OVOC samples (28
in 2007, 124 in 2013 and 124 in 2016) were collected and analysed in the three sampling
campaigns (Table S1).
**2.3 Chemical analysis**
2.3.1 Analysis of VOCs
The concentrations of 48 speciated non-methane hydrocarbons (NMHCs) in the canisters
were determined with an Entech Model 7100 Preconcentrator (Entech Instruments Inc.,
California, USA) coupling with a gas chromatography-mass selective detector (Model 5973N,
Agilent Technologies, USA), a flame ionization detector, and an electron capture detector
(GC-MSD/FID/ECD). The NMHCs were analysed in Donald Blake's laboratory at
University of California, Irvine (UCI) for the samples collected in 2007, in Guangzhou
Institute of Geochemistry (GIG), Chinese Academy of Sciences for the samples collected in
2013 and in The Hong Kong Polytechnic University (HKPolyU) for the samples collected in
2016. It should be noted that the GC-MSD/FID/ECD system in the latter two institutes was
the same as that at UCI, and inter-comparisons were performed regularly among the three
institutes, which showed reasonably good agreements (Ling et al., 2014; Wang et al., 2018b;



Zeng et al., 2018). Detailed information about the analysis procedures and quality assurance
and control can be found in Colman et al. (2001) and Simpson et al. (2010). Table S3
summarizes the limits of detection (LoDs), precisions and accuracies of the VOC analyses in
the three institutes.
The OVOC samples were stored in a refrigerator at 4 ℃ after sampling. For analyses of
OVOCs, the cartridges were eluted slowly with 2 ml of acetonitrile into a 2-ml volumetric
flask. A high-performance liquid chromatography (HPLC) system (Perkin Elmer Series 2000,
MA, USA) coupled with an ultraviolet (UV) detector operating at 360 nm was used for
analysis. The instrument was calibrated using standards of 5 gradient concentrations covering
the concentrations of interest for different OVOCs in ambient air. Good linear relationships
($R^2 > 0.999$) between the standard concentrations and responses of the instrument were
obtained for the 16 analysed OVOC species. The built-in computerized programs of quality
control systems such as auto-linearization and auto-calibration were used to guarantee the
data quality. Detailed information about the analysis and quality control of OVOC samples
was provided in Cheng et al. (2014), Cui et al. (2016) and Ling et al. (2016b). Due to the low
detection rate of many OVOCs, this study only focused on formaldehyde, acetaldehyde,
acetone and propionaldehyde, which had relatively high concentrations.
**2.4 Model description**
**2.4.1 Positive matrix factorization (PMF)**
PMF is a receptor model that has been extensively used for source apportionment of airborne
particulate matters and VOCs (Lee et al., 1999; Brown et al., 2007). In this study, US EPA
PMF 5.0 model (US EPA, 2017) was applied to identify the sources of $O_3$ precursors,
according to Equation (1) (Paatero, 1997; Ling et al., 2014).





$$x_{ij} = \sum_{k=1}^{p} g_{ik}f_{kj} + e_{ij} \qquad \text{Equation (1)}$$

where $x_{ij}$ is the measured concentration of $j$th species in $i$th sample, $g_{ik}$ represents the
contribution of $k$th source to $i$th sample, $f_{kj}$ denotes the fraction of $j$th species in $k$th source,
and $e_{ij}$ is the residual for $j$th species in $i$th sample. $p$ stands for the total number of
independent sources (Paatero, 2000a, b).
The uncertainties of the concentrations applied to PMF were set in the same way as Polissar
et al. (1998) and Reff et al. (2007). Values below or equal to the LoD were replaced by half
of the LoDs and the uncertainties for these values were set as 5/6 of the corresponding LoDs.
For the values greater than LoDs, the uncertainties were calculated as [(Error Fraction ×
concentration)$^2$ + (LoD)$^2$]$^{1/2}$ where 10% was assigned as the error fraction. Missing values
(mainly due to maintenance or malfunction of the instruments) were replaced by the
geometric mean of the measured values and their accompanying uncertainties were set as
four times the geometric mean value. More details about the settings of the uncertainty were
provided in Norris et al. (2008) and Zhang et al. (2012).
The model was run for 20 times with a random seed, and tests with different number of
factors were conducted. The optimum solution was finally determined based on both a good
fit to the observed data and the most reasonable and interpretable results according to the
knowledge on the sources of $O_3$ precursors in Hong Kong (Ling et al., 2011, 2014; Ou et al.,

253    2015).

**2.4.2 Observation-based model (OBM)**
A photochemical box model coupled with the Master Chemical Mechanism (PBM-MCM)
was used to simulate the photochemical $O_3$ formation on the VOC sampling days. In this
study, MCM v3.2, a near explicit chemical mechanism consisting of 5,900 species and



16,500 reactions which fully describes the homogeneous gas phase reactions in the
atmosphere (Jenkin et al., 1997, 2003; Saunders et al., 2003), was used. The observation data
of temperature, relative humidity, $O_3$, $SO_2$, CO, NO, $NO_2$ and 52 $C_2$-$C_{10}$ VOCs/OVOCs were
input into the model. Specifically, the 52 VOCs/OVOCs included 19 alkanes, 16 alkenes, 13
aromatics and 4 OVOCs, as shown in Table S4, where the statistics of the mixing ratios of
VOCs/OVOCs are also presented. Nitrous acid (HONO) was not monitored in this study. The
average diurnal cycle of HONO mixing ratios measured at the same site in autumn in 2011
(Xu et al., 2015) was input into the model to roughly represent its role in $O_3$ formation and
atmospheric radical chemistry. Due to the data limitation, the trends of HONO at TC in the
three sampling campaigns were not traceable. However, the measurements at a background
site in Hong Kong indicated comparable levels of HONO ($p>0.1$) between the autumn in
2012 and in 2018 (unpublished data). Therefore, adopting the HONO measured in 2011 as
the inputs of the simulations in the three sampling campaigns was likely a plausible
assumption, despite some uncertainties. The model was also tailored to the real situations in
Hong Kong. Specifically, the height of the planetary boundary layer was allowed to vary
from 300 m at night to 1400 m at noon. The photolysis rates were calculated according to the
measured solar radiations by the Tropospheric Ultraviolet and Visible Radiation model
(Madronich and Flocke, 1999; Wang et al., 2017a), with the detailed method described in
Lyu et al. (2017b). In addition to the chemical processes, the exchange between the lower
troposphere and free troposphere, and dry deposition were also considered in the model. The
concentrations of air pollutants in the free troposphere were set according to the observations
at a mountainous site in Hong Kong (Lam et al., 2013). The dry deposition rates were
adopted from the previous studies (Saunders et al., 2003; Lam et al., 2013). The other
physical processes were not included in the model, which might lead to insufficient
description of the transport. However, since the model was constrained to the observations



which included the transported air pollutants, the regional transport was partially considered.
Besides, the observations at 07:00 on each day were used to initiate each day's modelling,
through which the effect of regional transport before the daytime modelling was also
considered. We admit that the PBM-MCM cannot perfectly reproduce the real atmospheric
processes. However, it performed well in describing the in-situ photochemistry in previous
studies (Lam et al., 2013; Ling et al., 2014; Lyu et al., 2017b; Wang et al., 2017a). Actually,
the deficiency of PBM-MCM in consideration of the atmospheric dynamics enabled us to
assess the contributions of regional transport to $O_3$ in Hong Kong, based on the differences
between the observed and simulated $O_3$ (Wang et al., 2017a).
**2.5 Simulation scenarios**
Two scenarios of model simulation were performed in this study, *i.e.,* Scenario A and
Scenario B. The scenario A simulated the $O_3$ photochemistry in the whole air, which was
constrained by the observed concentrations of all the $O_3$ precursors. The model simulations in
scenario B (including six assumed sub-scenarios) were constrained by the concentrations of
$O_3$ precursors with those contributed by individual sources being subtracted from the
observed concentrations. Text S1 elaborates the set-up of these scenarios. The simulated $O_3$
in scenario A was regarded as the locally produced $O_3$, as the observed $O_3$ concentrations
were not input to constrain the model. Bearing in mind that the regional effects cannot be
completely eliminated in this approach, due to the impacts of regional air on the observed
concentrations of $O_3$ precursors. The differences between the scenario A and scenarios B
reflected the contributions of the individual sources to the simulated $O_3$ production rate.
**3 Results and discussion**





### 3.1 Observation overview


Figure 2 shows the hourly mixing ratios of $O_3$ observed at TC in the autumns of 2007-2017
with the data on VOC sampling days being highlighted in red. It was found that the autumn
$O_3$ increased significantly from 2007 to 2013 ($p<0.01$), with a rate of $0.34\pm0.002$ ppbv yr$^{-1}$.
This was consistent with Wang et al. (2017a) who reported an overall increase rate of autumn
$O_3$ of $0.67\pm0.07$ ppbv yr$^{-1}$ at the same site for the period of 2005-2013. On one hand, the
discrepancy in $O_3$ increasing rates might be due to the different statistics used to draw the
rates, *i.e.* hourly values in this study and monthly averages in Wang et al. (2017a). On the
other hand, the autumn $O_3$ increased substantially from $23.9\pm0.97$ ppbv in 2005 to $30.2\pm0.97$
ppbv in 2007, much quicker than the increase between 2007 and 2013. Without the inclusion
of the period of 2005-2007 might be another reason of the less $O_3$ enhancement calculated
here. In contrast to the increased autumn $O_3$ during 2007-2013, the autumn $O_3$ decreased
obviously from 2013 to 2017 ($p<0.01$), at a rate of $-2.27\pm0.003$ ppbv yr$^{-1}$, indicating a
fundamental alleviation of $O_3$ pollution in Hong Kong in the latest 5 years. Overall, a
statistically significant decreasing trend (rate = $-0.44\pm0.001$ ppbv yr$^{-1}$) was observed for the
autumn $O_3$ at TC through 2007 to 2017 ($p<0.05$). The average $O_3$ on VOC sampling days in
the three sampling campaigns also followed the same pattern, which increased from
$32.8\pm2.6$ ppbv in 2007 to $36.9\pm2.3$ ppbv in 2013, while decreased to $24.4\pm1.9$ ppbv in 2016.
Further, we investigated the number of $O_3$ episode days in the autumns of the three VOC
sampling years (see Figure S1) and identified 15 (16.5% of the autumn days, same below)
and 16 (17.6%) $O_3$ episode days in 2007 and 2013, respectively. However, there was only 5
(5.5%) $O_3$ episode days in the autumn of 2016. Similarly, the $O_3$ episode days accounted for
12.5%, 26.3% and 5.6% of the 2007, 2013 and 2016 sampling campaigns, respectively.
Therefore, the increase of $O_3$ from 2007 to 2013 and the decrease in the following years
could be represented by $O_3$ observed in the three sampling periods.





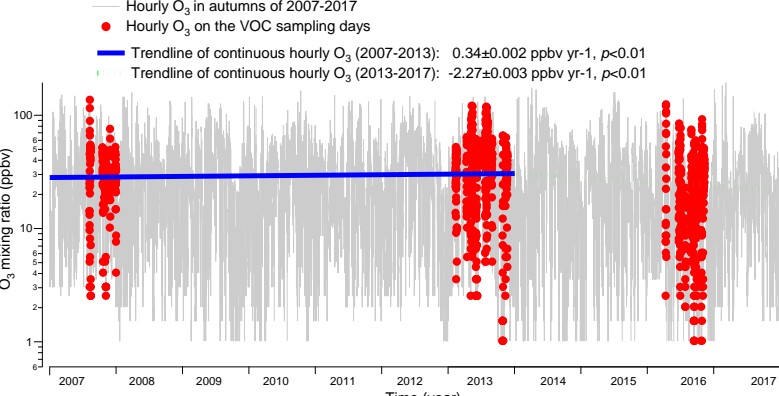

Figure 2. Long-term trends of the observed $O_3$ at TC from 2007 to 2017. Hourly $O_3$ values on the VOC sampling days in the autumns of 2007, 2013 and 2016 are marked in red.

Table 1 and Table S5 present the observed $O_3$, CO, NO, $NO_2$, $SO_2$ and TVOCs, as well as the meteorological conditions averaged on the VOC sampling days in 2007, 2013 and 2016, respectively. From 2007 to 2013, the TVOCs decreased by nearly a half, which was expected to result in the reduction of $O_3$ in view of the VOC-limited regime of $O_3$ formation at TC (Cheng et al., 2010; Wang et al., 2017a). However, the increases of CO and the notable decrease of NO in 2013 could enhance the $O_3$ production. The higher $O_3$ in 2013 indicated that this effect overrode the reduction of TVOCs in influencing the $O_3$ production. In particular, the decrease of NO meant the reduced NO titration to $O_3$, which has been recognized as a primary reason of $O_3$ increase in VOC-limited regime (Chou et al., 2006; Wang et al., 2018b). From 2013 to 2016, the decrease of $O_3$ was accompanied by the reductions of TVOCs and $NO_2$, though CO remained increasing at the same time. $NO_2$, as a direct source of $O_3$ through photolysis, plays important role in modulating the $O_3$ variation. Though the causes of $NO_2$ reduction are unknown to us, it might be one of the critical factors contributing to the decline of $O_3$ in Hong Kong in recent years. On the contrary, the increase





of CO was also confirmed by the continuous monitoring data at TC, with a rate of
$33.9 \pm 0.7$ ppbv yr$^{-1}$ between 2013 and 2016. In fact, the consistent increasing trend ($p<0.05$)
was also observed at the roadside sites in Hong Kong (not shown here). While the causes of
CO increase in Hong Kong may be complicated, the increased vehicle emission is a plausible
explanation. Studies (Johnson, 2008; Yao et al., 2008) revealed that while the new engine
technologies performed well in reducing $NO_x$ emission, they might lead to the increased
emission of CO, with the application of lower air-to-fuel ratio and engine temperature.
In addition, studies have confirmed that continental anticyclones and tropical cyclones are
conducive to severe $O_3$ pollution in Hong Kong, because these synoptic systems are often
accompanied with northerly winds, high temperature, strong solar radiation, and relatively
high pressure in Hong Kong (Ding et al., 2004; Huang et al., 2005; Jiang et al., 2015).
Table S6 summarizes number of $O_3$ episode days with tropical cyclone, continental
anticyclone and low pressure trough in the autumns of 2007, 2013 and 2016. In autumn 2007,
8, 8 and 1 $O_3$ episode day(s) were found to be related to the tropical cyclone, continental
anticyclone and low-pressure trough, respectively, with 2 $O_3$ episode days under the
combined influence of tropical cyclone and continental anticyclone. There were also 11 and 5
$O_3$ episode days in association with tropical cyclone and continental anticyclone in autumn
2013, respectively (Wang et al., 2018b). However, 4 of the 5 episode days found in autumn
2016 were associated with tropical cyclone, with the other one relative to low-pressure trough.
Therefore, the lower $O_3$ and less $O_3$ episode days in 2016 were also benefited from the
meteorological conditions.
Table 1. Mixing ratios of the measured trace gases and TVOCs averaged on the selective 45
VOC sampling days in 2007, 2013 and 2016.

| | 2007 | 2013 | 2016 |
| --- | --- | --- | --- |



| Unit: ppbv | Mean ±95% C.I. | Max. | Mean ±95% C.I. | Max. | Mean ±95% C.I. | Max. |
|---|---|---|---|---|---|---|
| $O_3$ | 32.8±2.6 | 137.0 | 36.9±2.2 | 121.2 | 24.4±1.9 | 124.9 |
| CO | 456.3±19.8 | 847.0 | 585.0±11.9 | 1047.9 | 691.8±9.5 | 1074.7 |
| NO | 17.2±3.2 | 124.7 | 10.9±1.3 | 98.6 | 11.3±1.4 | 94.6 |
| $NO_2$ | 27.7±2.1 | 69.6 | 31.5±1.4 | 80.8 | 22.0±1.1 | 103.2 |
| $SO_2$ | 6.9±0.4 | 21.8 | 7.0±0.2 | 18.0 | 3.0±0.1 | 10.7 |
| TVOCs | 49.7±4.4 | 111.1 | 25.1±1.4 | 68.0 | 21.1±1.4 | 71.9 |

**3.2 Model simulation of $O_3$**
**3.2.1 Model validation**
Figure 3 compares the simulated $O_3$ in scenario A and the observed $O_3$ on the VOC sampling
days. Overall, both the magnitudes and the temporal patterns of the observed $O_3$ were
reasonably reproduced, though the mean of the simulated $O_3$ (30.0±1.7 ppbv) was slightly
lower than the observed average (38.1±2.0 ppbv). To quantitatively evaluate the model
performance, the index of agreement (IOA) was used to examine the goodness of fit between
simulated and observed $O_3$. Within the range of 0-1, higher IOA represents better agreement
between the simulated and observed values (Willmott, 1982). In this study, the overall IOA
for the three sampling periods was 0.68, within the range of IOA (0.67-0.89) accepted by the
previous studies (Wang et al., 2015; Lyu et al., 2015, 2016a, c; Wang et al., 2017a, 2018a).
Good correlations ($R^2$=0.61) were also shown between the simulated and observed hourly $O_3$.
Bearing in mind the deficiencies of the box model in describing the atmospheric dynamics,
we believed that the modelling results were acceptable, but special attention and explanation
to the discrepancies between the simulated and observed $O_3$ was needed.





It was found that the discrepancies were most likely caused by the transport processes, *i.e.,*
vertical and horizontal transport, which were not fully represented in the PBM-MCM model
(George et al., 2013; Lakey et al., 2015; Wang et al., 2017a). For example, the simulated $O_3$
(maximum: 122.6 ppbv) was much higher than the observed $O_3$ (maximum: 44.3 ppbv) on
November 16, 2007, when the strong southeast winds (wind direction: 90°-180°) with the
highest wind speed of 5.3 m s$^{-1}$ prevailed in Hong Kong. The south sector winds from SCS
might dilute the locally produced $O_3$ and the $O_3$ precursors/intermediates (such as the radicals)
which were not constrained by the observations. The same circumstances were also observed
on October 27, November 17, 2007 and September 11-12, November 20, 2013, with
southeast winds dominated (74.4%) during the daytime (Figure 3). For those days with the
simulated $O_3$ lower than the observed $O_3$, *i.e.* October 3, 22-25, 2013 and November 6, 2016,
69.3% of the winds during the daytime came from the north (wind directions: 0-90° and
270°-360°), which might transport the air masses laden with $O_3$ and/or $O_3$
precursors/intermediates not constrained to the observations from inland PRD to the sampling
site. The observed $O_3$ mixing ratios are plotted against the wind fields in Figure S2. It is
obvious that $O_3$ were higher under the north winds, while lower in the south wind sectors,
confirming the effects of dilution and regional transport of the south and north winds on $O_3$
pollution in Hong Kong, respectively.





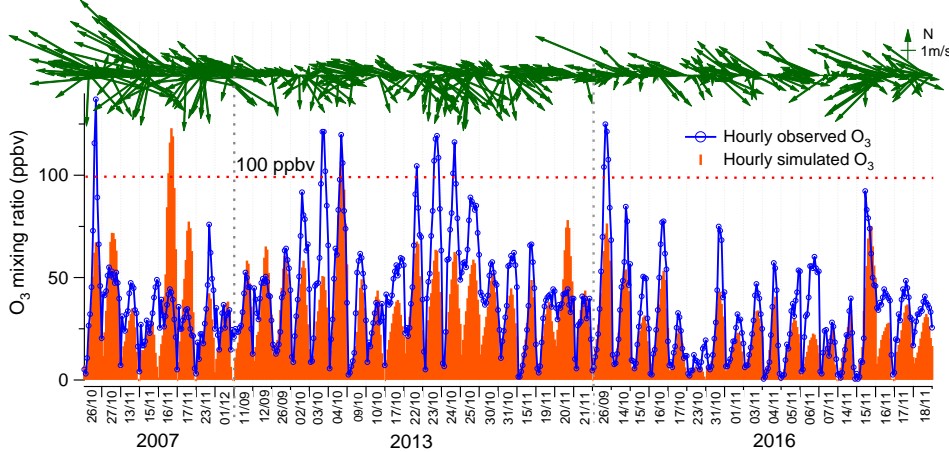


Figure 3. Hourly mixing ratio of the simulated and observed $O_3$ at TC during the VOC sampling periods in 2007, 2013 and 2016. The arrows represent the hourly wind sectors monitored at the sampling site.

**3.2.2 Inter-annual variations of the locally produced and regional transported $O_3$**

As discussed in section 2.5, the simulated $O_3$ in scenario A could be regarded as the locally produced $O_3$. Therefore, the differences between the observed $O_3$ and $O_3$ simulated in scenario A were treated as the regionally transported $O_3$ (Wang et al., 2017a). It is noteworthy that some negative values were generated with this method, corresponding to the dilution of the south winds to the locally produced $O_3$ as elaborated in section 3.2.1. Figure 4 shows the hourly mixing ratios of the observed, local and regional $O_3$ at TC in daytime hours (07:00-19:00 LT) of the three sampling campaigns. Overall, the observed $O_3$ was mainly (78.3±2.4%) contributed by the local photochemical production, with regional transport only accounting for 21.7±2.4% of the observed daily maximum $O_3$. However, regional transport was responsible for as high as 40.9±39.2% of the observed daily maximum $O_3$ in Hong Kong on the $O_3$ episode days when northerly winds prevailed, indicating the heavy $O_3$ burden superimposed by regional air masses from PRD. From 2007 to 2013, both the observed and





simulated locally-produced $O_3$ remained statistically unchanged ($p>0.1$), in contrast to the
increase of regional $O_3$ at a rate of 1.62±0.39 ppbv yr$^{-1}$ ($p<0.05$), similar to that
(1.09±0.21ppbv yr$^{-1}$) reported by Wang et al. (2017a) in the autumns of 2005-2013. However,
the decease of the locally produced $O_3$ in the same period as that simulated by Wang et al.
(2017a) was not seen here according to the simulated $O_3$ in the 2007 and 2013 sampling
campaigns. This discrepancy was likely caused by the limited samples in this study, no
OVOCs considered in Wang et al. (2017a) and/or the inexactly same study periods between
the two studies. Instead, we found that the locally produced $O_3$ showed a significant decline
at a rate of -4.35±0.10 ppbv yr$^{-1}$ during 2013-2016 ($p<0.05$), when the regionally transported
$O_3$ also decreased (10.9±2.0 and 8.9±1.8 ppbv in the 2013 and 2016 sampling campaign,
respectively). As a result, the increase of the observed $O_3$ from 2007 to 2013 was reversed by
the decrease between 2013 and 2016, leading to an overall decreasing trend of the observed
$O_3$ during 2007-2016 (rate = -0.54±0.15 ppbv yr$^{-1}$, $p<0.05$).





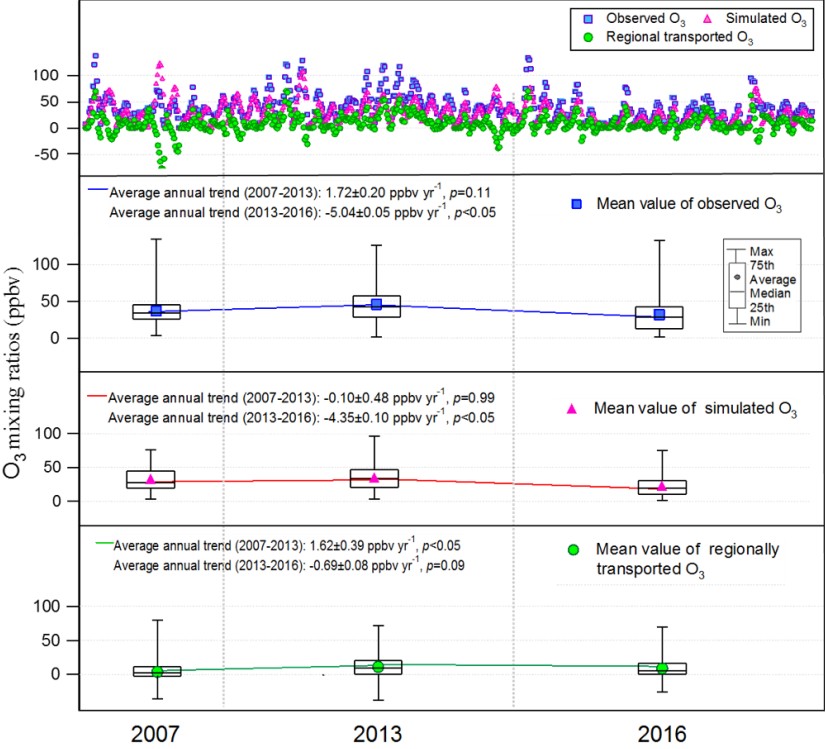


Figure 4. Hourly values (first panel) and the throughout-campaign statistical results (second
to fourth panels) of the observed, simulated (locally-produced) and regional $O_3$ mixing ratios
in daytime hours (07:00 – 19:00 LT) in the three sampling campaigns.

The significant alleviation of $O_3$ pollution in Hong Kong from 2013 to 2016 might be related
to the measures taken to control the emissions of $O_3$ precursors in Hong Kong and in
mainland China. The effectiveness of the actions launched by Hong Kong government in $O_3$
abatement was fully demonstrated in previous studies (Xue et al., 2014a; Lyu et al., 2017a;
Wang et al., 2017a), and would be further evaluated in this study (section 3.4). Besides, the
emission controls in mainland China might contribute to the decrease of $O_3$ in this period. For
example, the China's $NO_x$ emissions for the first time showed a decreasing trend from 2013,
benefited from the implementation of the China's Clean Air Action Plan (Zheng et al., 2018).





Furthermore, we looked into the monthly average $O_3$ observed at the 12 air quality
monitoring stations across the inland PRD, including three regional monitoring stations, *i.e.*
Tianhu, Wanqingsha and Jinguowan, and nine urban monitoring stations, *i.e.* Xiapu, Jinjuzui,
Donghu,    Tangjia,    Liyuan,    Huijingcheng,    Zimaling,    Luhu    and    Chengzhong
(https://www.epd.gov.hk /epd/sc_chi/resources_pub/publications/m_report.html).    As shown
in Figure 5, $O_3$ at these stations remained relatively stable (*p*=0.99) during 2006-2013, which
however showed a contrastively decreasing trend at a rate of -1.73±0.08 ppbv yr$^{-1}$ from 2013
to 2016. This corroborated our modelling results that the regional contribution to $O_3$ in Hong
Kong ceased increasing or even began to decrease since 2013. Though the substantial
decrease of $NO_x$ was a plausible reason for the alleviated regional $O_3$ pollution, analyses of
the causes are out of the scope of this study. In addition to the reduced local formation and
regional transport of $O_3$, the more favourable meteorological conditions in 2016 might be
another reason of the $O_3$ decrease, as discussed in section 3.1.

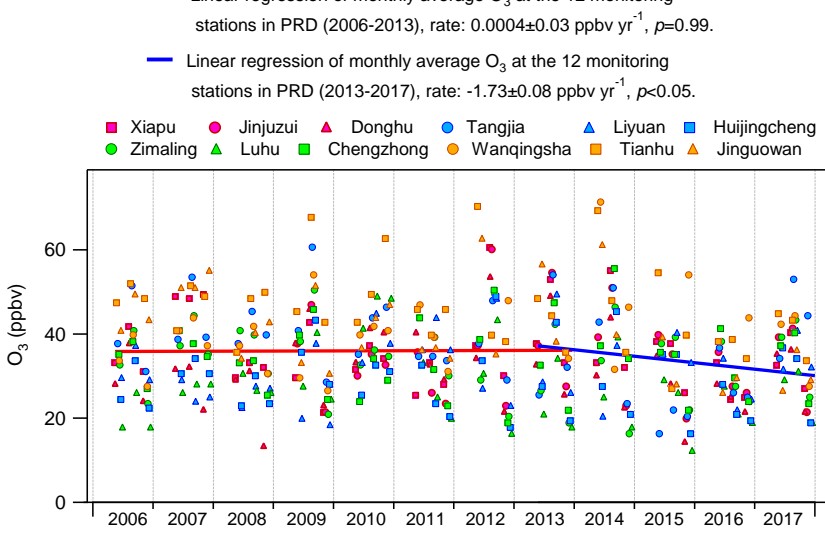


Figure 5. Trends of the observed monthly average $O_3$ at the 12 air quality monitoring stations
in inland PRD.




### 3.3 Local production and destruction pathways of $O_3$ and OH radical

### 3.3.1 In-situ net $O_3$ production

Figure 6 shows the average diurnal profiles of the simulated $O_3$ production and destruction pathways during the three sampling campaigns. Also shown are the average diurnal cycles of the simulated $O_3$. The shift of the peaks between the net $O_3$ production rate and the simulated $O_3$ was due to the accumulation of the newly generated $O_3$ over time in the model, which was also true in the real situations. The reactions between $NO_2$ and $O_3$, leading to the formation of $NO_3$ and $N_2O_5$, in addition to dry deposition and aloft exchange, were the main depletions of the simulated $O_3$ in the late afternoon. Consistent with previous studies (Kanaya et al., 2009; Liu et al., 2012; Xue et al., 2014b), these pathways were not included in the calculation of the net $O_3$ production rate, because we mainly focused on the photochemical processes in the hours when $O_3$ was accumulated. It was found that the reaction between $HO_2$ with NO dominated the $O_3$ production rates in all the cases, with an average rate of $3.0\pm0.6$ ppbv h$^{-1}$ ($72.6\pm15.1\%$, percentage of the total $O_3$ production rate, same below), $2.1\pm0.3$ ppbv h$^{-1}$ ($77.8\pm9.9\%$) and $1.1\pm0.2$ ppbv h$^{-1}$ ($80.7\pm11.6\%$) in the 2007, 2013 and 2016 sampling campaigns, respectively. In addition, the sum of the reaction rates between $RO_2$ radicals and NO contributed $1.1\pm0.2$ ppbv h$^{-1}$ ($27.4\pm5.3\%$), $0.6\pm0.1$ ppbv h$^{-1}$ ($22.2\pm3.0\%$) and $0.3\pm0.04$ ppbv h$^{-1}$ ($19.3\pm3.1\%$) to the $O_3$ production rate in 2007, 2013 and 2016, respectively. The formation of $HNO_3$ though the reaction between OH and $NO_2$ served as the main scavenger pathway of $O_3$, as $NO_2$ would be photolyzed and produce $O_3$ otherwise. On average, $O_3$ was consumed in this way at a rate of $-1.0\pm0.1$ ppbv h$^{-1}$ ($82.6\pm12.0\%$, percentage of the total $O_3$ destruction rate, same below), $-0.9\pm0.1$ ppbv h$^{-1}$ ($83.1\pm7.7\%$) and $-0.5\pm0.05$ ppbv h$^{-1}$ ($83.4\pm8.4\%$) in 2007, 2013 and 2016, respectively. The photolysis of $O_3$ was the second contributor to $O_3$ destruction, with an average contribution of $-0.09\pm0.01$ ppbv h$^{-1}$ ($10.5\pm1.1\%$) for the three sampling periods. Besides, the ozonolysis of unsaturated



VOCs and the reactions between $O_3$ and radicals (OH and $HO_2$) were responsible for 4.1±0.5%
and 2.3±0.3% of the total destruction rate of the locally produced $O_3$, respectively.
Overall, the net local $O_3$ production rate decreased from 3.0±0.7 ppbv $h^{-1}$ in 2007, to
1.6±0.3 ppbv $h^{-1}$ in 2013, till 0.8±0.2 ppbv $h^{-1}$ in 2016, corresponding to the decline of the
locally produced $O_3$ through 2007 to 2016 (Section 3.2.2).

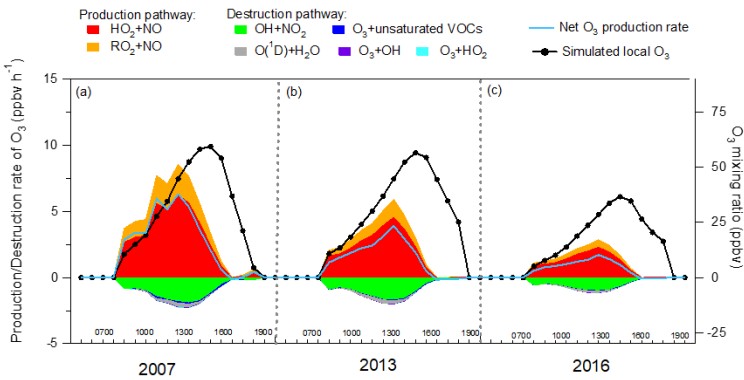


Figure 6. Average diurnal profiles of the local $O_3$ production and destruction rates in the
sampling campaigns of (a) 2007, (b) 2013 and (c) 2016.
**3.3.2 Recycling of OH radical**
As one of the most important radicals in the atmosphere, OH initiates the oxidation of VOCs,
leading to $O_3$ formation. Figure 7 presents the average diurnal profiles of the simulated OH
and the formation and loss pathways dominating the recycling of OH during the three
sampling periods. According to the simulation by PBM-MCM model, the average OH
concentration was (1.4±0.3) × $10^6$ molecules $cm^{-3}$ in the 2007 sampling campaign, which
significantly decreased to (1.3±0.2) × $10^6$ molecules $cm^{-3}$ in the 2013 sampling campaign
($p<0.05$) and further decreased to (0.9±0.1) × $10^6$ molecules $cm^{-3}$ in 2016 ($p<0.05$).
As expected, the formation and loss rates of OH were basically balanced in all the cases. OH
was mainly formed from the reaction of $HO_2+NO$, which accounted for 58.2±2.3% of the





total OH production rate over the three sampling campaigns. The photolysis of HONO and
O$_3$ made comparable but much lower contributions (19.2±1.4% and 19.3±2.9%, respectively)
to the production of OH, with the rest attributable to the ozonolysis of unsaturated VOCs
(2.8±0.2%) and the photolysis of H$_2$O$_2$ (0.2±0.01%). On the contrary, OH was mainly
depleted by the reactions with NO$_2$ (39.2±1.1%), VOCs (25.3±0.9%), CO (21.0±0.6%) and
NO (14.1±1.1%).
Consistent with the variations of the local O$_3$ production, both the local formation and loss
rates of OH decreased through 2007 to 2016 ($p<0.05$), with much more obvious reductions in
the later phase (2013-2016). On one hand, the continuous reduction of VOCs resulted in
lower HO$_2$ and RO$_2$ concentrations (Figure S3), hence the lower production rate of OH
through the reaction of HO$_2$+NO. At the same time, the destruction rates of OH also
decreased due to the reductions of OH and the O$_3$ precursors, except for CO (Figure 7 and
Table 1). The decreases of the OH production and destruction rates indicated that the
propagation of the reaction cycles, namely the recycling of OH, became slower from 2007 to
2016. This also explained why the locally produced O$_3$ decreased in these ten years, since O$_3$
is formed with the consumption and recycling of OH radical.

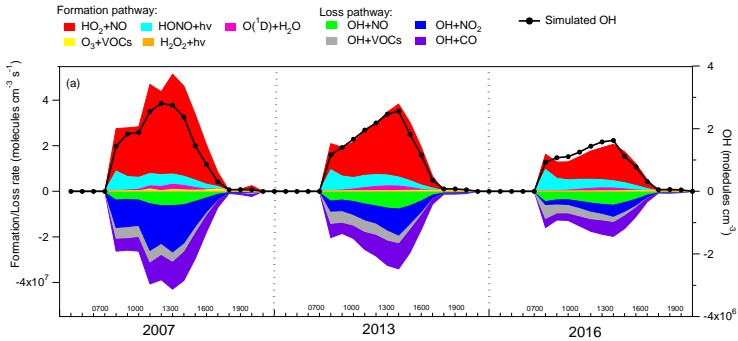


Figure 7. Average diurnal cycles of the OH formation and loss rates during the sampling
periods in (a) 2007, (b) 2013 and (c) 2016.



### 3.4 Source contributions to the production of O$_3$ and radicals

### 3.4.1 Source apportionment

To resolve the sources of O$_3$ precursors, 27 species, including CO, NO, NO$_2$, 12 alkanes, 4 alkenes and 8 aromatics, were applied to PMF for source apportionment. These species were either of high abundances or typical tracers of VOC sources in Hong Kong. Source apportionment was conducted for a total of 414 samples covering the three sampling periods, so that the uncertainty of the source apportionment results could be reduced, compared to separate source apportionments for each of the three sampling periods. Figure 8 shows the average profiles of the six sources resolved by PMF. The modelling errors were estimated with the bootstrap method integrated in PMF (Brown et al., 2015).

Factor 1 was assigned as the combination of LPG usage and gasoline evaporation, in view of the high loadings of C$_2$-C$_5$ hydrocarbons. Specifically, propane and *i-/n*-butanes are the main components of LPG in Hong Kong, and gasoline evaporation generally contains large quantities of *i-/n*-pentanes, in particularly *i*-pentane (Guo et al., 2013; Lyu et al., 2017a). Factor 2 was characterized by moderate to high percentages of *i-/n*-pentanes and TEX (toluene, ethylbenzene and xylenes). These species are commonly seen in gasoline exhausts. Therefore, we defined this factor as gasoline exhausts. Both the third and fourth factors indicated solvent-related emissions. While Factor 3 likely represented household solvent usage, due to the dominance of hexane and hexane isomer (3-methylpentane) (Ling and Guo, 2014; Ou et al., 2015), Factor 4 was more related to emissions from coatings and paints, in view of the dominance of the aromatics (Ling and Guo, 2014). Factor 5 was distinguished by the high concentrations of ethane, ethene, ethyne and benzene, together with the relatively heavy (C$_7$-C$_{10}$) alkanes, which are typical species in diesel exhausts (Schauer et al., 1999; Kashdan et al., 2008; Sahoo et al., 2011). Therefore, this factor was designated as diesel



exhausts. The last factor denoted for biogenic emissions (BVOCs), due to the exclusive
dominance of isoprene (Guenther, 2006).

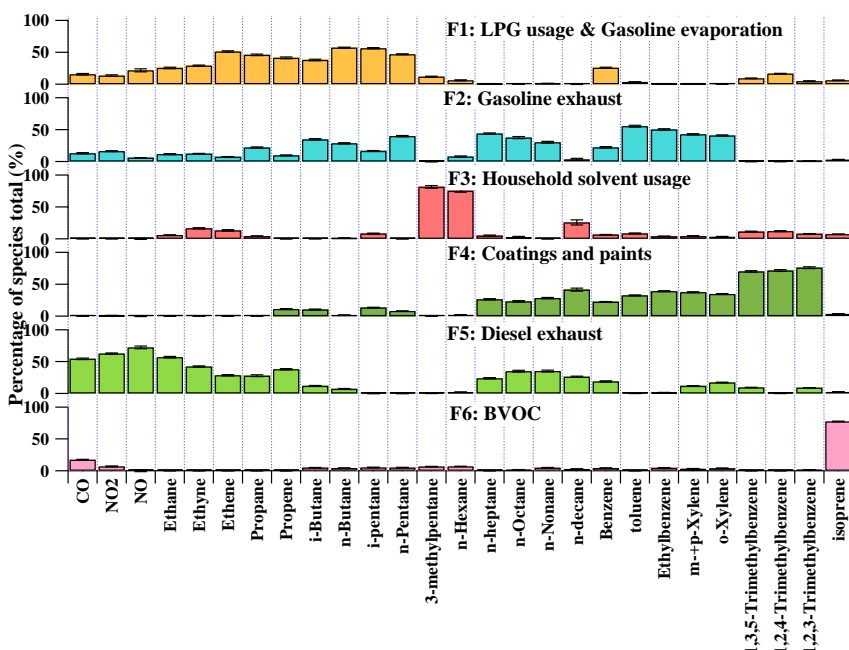


Figure 8. Average profiles of the $O_3$ precursors sources at TC in the three sampling
campaigns. The uncertainties were estimated with the bootstrap method in PMF.

Figure S4 presents the total mixing ratio of VOCs emitted from each individual source
extracted from PMF during the three sampling periods in Hong Kong. The VOC emissions
from LPG usage and gasoline evaporation decreased significantly ($p<0.05$) at a rate of -
2.61±0.03 ppbv $yr^{-1}$ from 2007 to 2016. However, the VOCs in association with gasoline
exhausts experienced an increase (rate = 1.32±0.02 ppbv $yr^{-1}$, $p<0.05$) in these years,
indicating that the reduction of VOC emissions from LPG usage and gasoline evaporation
was not attributable to the change in emissions of gasoline-fuelled vehicles. Insight into the
mixing ratios of propane and *i-/n*-butanes (LPG tracers) in this source revealed a significant





decline from 3.51±0.52 ppbv in the 2007 sampling campaign to 1.27±0.11 ppbv in the 2016
sampling campaign. Therefore, the reduction of VOC emissions from LPG usage was most
likely the reason of the decrease of VOCs allocated to the source of LPG usage and gasoline
evaporation. In fact, it was confirmed by our previous studies (Lyu et al., 2016b; Yao et al.,
2019) that the replacement of catalytic converters on LPG-fuelled vehicles during September
2013-May 2014 effectively reduced the VOC emissions from LPG-fuelled vehicles in Hong
Kong. In addition, the variations in LPG usage in inland PRD, where LPG was extensively
used as vehicular and domestic fuels (Liu et al., 2008), might also contribute to the emission
reduction of VOCs, in view of the decrease of LPG tracers in this source from 2007
(3.51±0.52 ppbv) to 2013 (2.04±0.27 ppbv), when no control was performed against LPG
fuelled vehicle emissions in Hong Kong. The VOCs emitted from solvent usage (including
the household solvent, coatings and paints) also decreased significantly ($p<0.05$) from 2007
to 2016, likely benefiting from the actions taken to restrict the VOC contents in solvent
products starting from 2007 (phase I) and 2010 (phase II) in Hong Kong (Lyu et al., 2017a).
VOCs attributable to diesel exhausts decreased ($p<0.05$) from the 2007 (2.6±0.3 ppbv) to
2013 sampling campaign (2.0±0.2 ppbv), which however were unchanged between 2013 and
2016 (2.2±0.2 ppbv). In fact, a subsidy program has been implemented in Hong Kong since
2007 to progressively eliminate the pre-Euro IV diesel vehicles or to upgrade their emission
standards to Euro IV (HKEPD, 2017b), and the effectiveness of this program in VOC
reductions till 2013 was confirmed by Lyu et al. (2017a) with the online measurement data at
the same site. However, while the phase III of this program (2014-2019) is still ongoing, the
VOCs emitted from diesel vehicles remained stable between the 2013 and 2016 sampling
campaigns. This undesirable result might be due to the fact that the actions were mainly
targeted at the pre-Euro, Euro I and Euro II diesel vehicles before 2013, whereas the phase III
of the program initiated in 2014 focused on the Euro III vehicles (HKEPD, 2017b, 2018).



Since the former were vehicles with higher emissions, it is not unreasonable that reduction of
VOCs was more discernible between 2007 and 2013. Further, the effectiveness of the phase
III program might be somewhat offset by the wearing-out of the pre-existing vehicles and the
increase of diesel vehicle populations (Competition Commission, 2017). Further evaluation
with more data in a longer period is recommended. At last, the increase of BVOCs from 2007
to 2013 but comparable levels between 2013 and 2016 seemed to be related to the lower
($p<0.05$) temperature in the 2007 sampling campaign (Table S5).
**3.4.2 Source contributions to $O_3$ production**
Figure 9 presents the contributions of VOCs emitted from individual sources to the
production and destruction rates of $O_3$, as well as the simulated contributions to the $O_3$
mixing ratios. $NO_x$ was not included in these analyses, because of its relatively high
uncertainties in source apportionment results due to the short lifetimes. Consistent with the
$O_3$ production and destruction in the whole air, the pathway of $HO_2+NO$ dominated over the
reactions between $RO_2$ and NO in $O_3$ production for all the individual sources. The
destruction of $O_3$ was mainly driven by $NO_2$ reacting with OH. For the net $O_3$ production rate,
VOCs attributable to the coatings and paints made the largest contribution ($0.38\pm0.05$ ppbv h$^-$
$^1$), followed by gasoline exhausts ($0.22\pm0.03$ppbv h$^{-1}$), LPG and gasoline evaporation
($0.21\pm0.03$ppbv h$^{-1}$), BVOCs ($0.19\pm0.03$ppbv h$^{-1}$), household solvent usage ($0.15\pm0.04$ ppbv
h$^{-1}$) and diesel exhausts ($0.13\pm0.01$ppbv h$^{-1}$). Despite some peak shifts for the reasons
illustrated in section 3.3.1, the $O_3$ mixing ratios elevated by the individual sources followed
the same pattern as the net $O_3$ production rates, with the highest $O_3$ enhancement ($1.92\pm0.21$
ppbv) by the source of coatings and paints and the lowest increase by household solvent
usage ($0.86\pm0.06$ ppbv) and diesel exhausts ($0.83\pm0.06$ ppbv). The contributions of source-
specific VOCs to $O_3$ production, particularly the importance of solvent usage in $O_3$ formation
in Hong Kong, were generally in line with previous studies (Ling and Guo, 2014; Ou et al.,



2015). This was actually expected according to the reactivity of major VOCs in each source.
For example, the TEX in the source of coatings and paints (Figure 8) have been identified to
be of high $O_3$ formation potentials (Lau et al., 2010; Ling et al., 2011, 2013). However, the
PBM-MCM model simulations enabled us to quantitatively evaluate the contributions of
VOC sources to $O_3$ production rates.

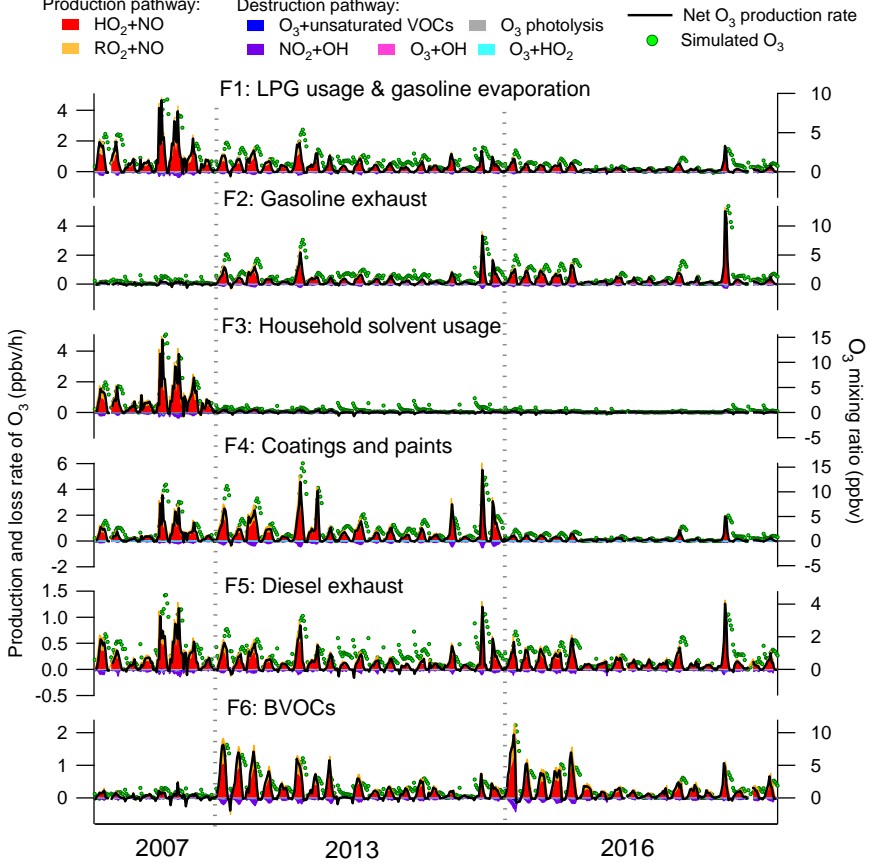


Figure 9. Contributions of VOCs in individual sources to the production and destruction rates
of $O_3$ and to the $O_3$ mixing ratios in the three sampling campaigns.
From a historical perspective, we found that the contribution of LPG usage and gasoline
evaporation to $O_3$ production significantly decreased ($p<0.05$) from 2007 to 2016 sampling




623 campaign (2007: 0.51±0.11 ppbv h$^{-1}$; 2013: 0.20±0.03 ppbv h$^{-1}$; 2016: 0.10±0.02 ppbv h$^{-1}$),

624 which coincided with the variations of VOCs emitted from LPG-fuelled vehicles as discussed

625 above. Gasoline exhaust contributed much less ($p$<0.05) to the net $O_3$ production rate in 2007

626 (0.02±0.01ppbv h$^{-1}$), than those in 2013 (0.26±0.05ppbv h$^{-1}$) and 2016 (0.27±0.07ppbv h$^{-1}$),

627 in line with the variations of VOCs emitted from this source. The reductions of VOC

628 emissions from solvents also resulted in the consistent decrease of the net $O_3$ production rate

629 from 1.22±0.17 ppbv h$^{-1}$ in the 2007 to 0.14±0.05 ppbv h$^{-1}$ in the 2016 sampling campaign.

630 The $O_3$ production rates contributed by VOCs in diesel exhausts were reduced from 2007

631 (0.21±0.05 ppbv h$^{-1}$) to 2013 (0.11±0.02 ppbv h$^{-1}$) and remained unchanged thereafter (2016:

632 0.11±0.02 ppbv/h). The $O_3$ production rate traceable to BVOCs showed a significant increase

633 from 2007 (0.04±0.02 ppbv h$^{-1}$) to 2016 (0.22±0.04 ppbv h$^{-1}$), since the mixing ratios of

634 BVOCs significantly increased ($p$<0.05) in these years. It is noteworthy that the changes in

635 meteorological conditions in these three sampling campaigns might also partially account for

636 the variations in the source contributions to $O_3$ production. For example, the 2013 sampling

637 campaign was characterized by the relatively higher temperature and lowest relative humidity

638 among the three sampling periods, which favoured $O_3$ formation in 2013 (Table S5). Besides,

639 due to limited samples in this study, we recommend further assessments with more data in

640 longer periods to be carried out in future study.

641 **4 Conclusions**

642 Photochemical pollution with high and increasing concentrations of $O_3$ has been an important

643 environmental issue in South China. With the observation data of $O_3$ and its precursors at a

644 suburban site in Hong Kong, downwind of South China, this study analysed the inter-annual

645 variations of $O_3$ and its photochemistry, as well as the contributions of VOC sources to the

646 local $O_3$ production rates in 2007, 2013 and 2016. To our knowledge, this is the first time that

647 a substantial alleviation of $O_3$ pollution in this region was identified between 2013 and 2016,





in contrast to the repeatedly confirmed $O_3$ increase before 2013. In addition to the changes in
meteorological conditions among the three sampling campaigns, the termination of the rise in
regionally transported $O_3$ and the decrease of the local $O_3$ production rate contributed to the
decline of $O_3$ in the later period. The emission reductions (particularly for $NO_x$) in mainland
China starting from 2013, the year when the China's Clean Air Act Plan was launched, might
more or less play a role in ceasing the increase of regional $O_3$. In Hong Kong, the
replacement of catalytic converters and the constraints of VOC contents in solvent products
led to the reductions of VOC emissions from LPG-fuelled vehicles and solvent usage,
respectively. As a result, the local $O_3$ production rate and the recycling rate of OH radical
decreased substantially from 2013 to 2016. Though the variations in meteorological
conditions and the limited sample size might somewhat introduce uncertainties to the
conclusions drawn from the present study, it is plausible that the local and regional
interventions were effective on the control of $O_3$ pollution in Hong Kong. Nevertheless,
studies with more data in longer periods should be conducted, not only in Hong Kong but
also in mainland China where $O_3$ is still increasing in most of the territories.
**Author contribution**
Hai Guo and Fei Jiang initiated and designed the experiments, and Xufei Liu and Xiaopu Lyu
carried them out. Xiaopu Lyu and Yu Wang developed the model code and performed the
simulations. Xufei Liu and Xiaopu Lyu prepared the manuscript and Hai Guo finalized the
manuscript with contributions from all co-authors.
**Acknowledgements**
This study was supported by the National Key R&D Program of China via grant No.
2017YFC0212001, Research Grants Council of the Hong Kong Special Administrative
Region Government via grants PolyU 152052/14E, PolyU 152052/16E and CRF/C5004-15E,





the Public Policy Research Funding Scheme from Policy Innovation and Co-ordination
Office of the Hong Kong Special Administrative Region Government (Project Number:
2017.A6.094.17D), and the Hong Kong Polytechnic University Ph.D. scholarships via
research project #RUDC.

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
