# Peer review of "Inter-comparison of O3 formation and radical chemistry in the past decade at a suburban site in"

_Atmospheric Chemistry and Physics, 2018_

## Referee Comment (RC1) · Anonymous Referee #3 · 8 Jan 2019

The paper describes a monitoring study of trace gases at a suburban site close to the airport in Hong Kong. A comparison is made of ozone concentrations over a ten year period via detailed analysis of three periods, coupled with master chemical mechanism / box model calculations to facilitate interpretation. This analysis is coupled with a source apportionment analysis for VOCs and with discussion of policy changes in the area. The study is one of a sequence of detailed papers that have investigated air quality, and especially ozone production, in this area, especially in relation to air quality policy. This paper adds to that corpus of knowledge and understanding and should be published. The authors should consider the following points:

[Figure]

1.The Introduction gives references to the large series of investigations of which this paper is the latest. It is important to understand what these papers have contributed to our understanding of air quality in Hong Kong, and information is provided, but it is quite difficult to appreciate fully. It would be helpful if a table was provided, listing the main contributions (not all need be included and there are some overlaps) of these papers: target pollutant, nature of the monitoring, main conclusions.

2.The main aim of the paper is to investigate the changes in ozone through measurement of the major inorganic species, including ozone, and a wide range of VOCs, at a single site. The observations are supported and interpreted using a box model based on the master chemical mechanism. There are several issues that warrant further discussion than they are given in the text, and some clarification:

i)The site (Tung Chung, TC) is close to a highway serving the airport which is only 3 km away. The mean NO and NO2 concentrations are not too high, but the maximum values are. It is commented that one reason for low [O3] could be titration by NO. An analysis of the degree of titration would be helpful.

ii)The paper is based on analysis of three periods covering 2007 – 2016. There can be significant year to year variations resulting, for example, from meteorological rather than AQ policies. Figure 2 shows that long term trends in hourly ozone at TC and shows that decreases have, on average, occurred since 2013. The figure is very difficult to interpret, though, since it shows log[O3]. The low concentrations, presumably at night, are of little interest. It would be much easier to understand what is happening if this plot were linear in [O3].

iii) Some further discussion of the locally produced and regionally transported ozone (p5 line 109) would be helpful. The wind speeds are generally quite low, so the distances the precursors and ozone itself are transported is not large. How significant is precursor transport? Are the sources greater in some directions for the longer-lived precursors? How significant is this? The assertion that the regionally transported emissions can be determined by difference as asserted on p5 needs some moderation – the chemistry isn't linear and the simulations aren't 100% accurate, as demonstrated by some apparently negative regional sources.

iv)HONO was not monitored in 2016 and the 2011 values were used. Some justification, based on measurements elsewhere, is provided. Figure 7 shows the OH formation and loss rates and demonstrates the importance of HONO in the early morning, especially in 2016. The potential error in this rate analysis should be emphasised.

v) This last point illustrates one of the deficiencies of the paper – the tendency to discuss the behaviour of average values and reaction rates rather than delve into the diurnal variations, which show some interesting effects, some difficult to understand. For example, in Fig 7, the high NO + HO2 rate shows that the chain length is quite long, especially in 2007 – the propagation rate considerably exceeds the initiation rates (mainly O3 + alkene and O1D + H2O). The OH production from HONO is largely balanced by HONO formation from OH + NO except in the early morning. On the other hand, the termination rate (OH + NO2) is quite large and significantly exceeds the initiation rates. Why is this? It needs some explanation. The effects are still there in the later years, although less pronounced.

vi) It would help understand these arguments, and those given on page 25, if Fig S3 were shown, and discussed in more detail, in the main text. Why do the concentrations of HO2 and RO2 increase in the late afternoon in 2016?

vii)The concentrations of oxygenates are very high (Table S4), yet their sources are not discussed in the main text. Are they produced by reaction or are there emission sources? Were the concentrations constrained, or were they simulated. Were high concentrations of other oxygenates generated in the simulations?

viii) Why were the BVOC concentrations so low in 2007 (Fig S4)? The temperature was slightly lower, but not by much, and the solar irradiance was higher (Table S5)

[Figure]

ix)While the analysis of the impact on O3 of the emission changes, via the scenario B simulations, is of considerable value, it is important, once again, to point out that the chemistry is non-linear and the approach limits the quantitative, but not the qualitative, nature of the conclusions.

x) Small point: the decrease in [OH] of less than 10% between 2007 and 2013 could hardly be described as significant (p24, line 502)

Some of these comments require relatively detailed responses, which may need to be consigned to the Supplement to avoid over complication of the main text. It is important, though, that the issues related to the detailed chemistry are resolved since much of the emphasis of the paper depends on the box model calculations.
* * *

---

## Referee Comment (RC2) · Anonymous Referee #2 · 13 Feb 2019

The authors have addressed my comments that I have provided in the initial assessment of the paper.

---

## Author Comment (AC2) · 22 Mar 2019

We thank the referee for his/her review.

---

## Author Response (AR1)

Responses to Referee #1

The paper describes a monitoring study of trace gases at a suburban site close to the airport in Hong Kong. A comparison is made of ozone concentrations over a ten-year period via detailed analysis of three periods, coupled with master chemical mechanism/box model calculations to facilitate interpretation. This analysis is coupled with a source apportionment analysis for VOCs and with discussion of policy changes in the area. The study is one of a sequence of detailed papers that have investigated air quality, and especially ozone production, in this area, especially in relation to air quality policy. This paper adds to that corpus of knowledge and understanding and should be published. The authors should consider the following points:

We highly appreciate the reviewer for the comments and suggestions. More detailed information is supplemented in the revised manuscript according to the reviewer's concerns.

1.  The introduction gives references to the large series of investigations of which this paper is the latest. It is important to understand what these papers have contributed to our understanding of air quality in Hong Kong, and information is provided, but it is quite difficult to appreciate fully. It would be helpful if a table was provided, listing the main contributions (not all need be included and there are some overlaps) of these papers: target pollutant, nature of the monitoring, main conclusions.

Thank you for this good comment. Table S1, as suggested, is provided in the revised manuscript supplementary, which lists the contributions of previous papers to the understanding on $O_3$ pollution in Hong Kong.

Revisions have been made as follows.

Overall, the previous studies have greatly enhanced our understanding on $O_3$ pollution in Hong Kong, and details about the studies can be referred to in Table S1.

For details, please refer to lines 87-89, page 4 and Table S1.

2. The main aim of the paper is to investigate the changes in ozone through measurement of the major inorganic species, including ozone, and a wide range of VOCs, at a single site. The observations are supported and interpreted using a box model based on the master chemical mechanism. There are several issues that warrant further discussion than they are given in the text, and some clarifications:

i) The site (Tung Chung, TC) is close to a highway serving the airport which is only 3 km away. The mean NO and $NO_2$ concentrations are not too high, but the maximum values are. It is commented that one reason for low [O3] could be titration by NO. An analysis of the degree of titration would be helpful.

Thank you for the good suggestion.

Located in a newly-developed town in western Hong Kong, TC site is expected to be influenced by local vehicle emissions and emissions from the nearby airport highway (Wang et al., 2018b). In TC, the daytime NO peaks usually occur at around 8:00 and 19:00 LT, attributable to the vehicular emissions in the morning and evening rush hours. Therefore, in the early morning (7:00 – 8:00 LT), the nocturnal steady-state $O_3$ is titrated by the freshly emitted NO, forming a trough in the diurnal cycle of $O_3$. Here, the decrease of $O_3$ in the early morning is employed to represent the degree of NO titration to $O_3$, $\Delta O_3$-1 in Figure S1 taking the diurnal cycle of $O_3$ in 2007 as an example. Correspondingly, $\Delta O_3$-2, the difference between the maximum $O_3$ in the afternoon and the minimum $O_3$ in the morning, reflects the photochemical production of $O_3$ in daytime. According to Figure S1, the $O_3$ titrated by NO ($\Delta O_3$-1) was 9.2±2.1, 9.8±2.7 and 6.8±2.7 ppbv, equivalent to 29.9±8.8%, 26.7±12.7% and 32.5±16.6% of the photochemically formed $O_3$ ($\Delta O_3$-2) in 2007, 2013 and 2016, respectively.

The discussions on the degree of titration have been added into the revised manuscript.

It was expected that $O_3$ at TC would be significantly influenced by NO emitted from the aforementioned sources. As shown in Figure S1, the $O_3$ titrated by NO ($\Delta O_3$-1) was equivalent to 29.9±8.8%, 26.7±12.7% and 32.5±16.6% of the photochemically formed $O_3$ ($\Delta O_3$-2) in the 2007, 2013 and 2016 sampling campaigns, respectively, confirming the importance of NO titration in modulating $O_3$ at the site.

[Figure]

Figure S1. Average diurnal cycles of $O_3$ in the 2007, 2013 and 2016 sampling campaigns. $\Delta O_3$-1: $O_3$ decrease in the early morning driven by NO titration; $\Delta O_3$-2: photochemically formed $O_3$ in the daytime (diurnal cycle of $O_3$ in 2007 as an example).

For details, please refer to lines 140-145, page 6 and Figure S1.

ii) The paper is based on analysis of three periods covering 2007-2016. There can be significant year to year variations resulting, for example, form meteorological rather than AQ policies. Figure 2 shows that long term trends in hourly ozone at TC and shows that decreases have, on average, occurred since 2013. The figure is very difficult to interpret, though, since it shows log[$O_3$]. The low concentrations, presumably at night, are of little interest. It would be much easier to understand what is happening if this plot were linear in [$O_3$].

Thanks for this good comment and the suggestion. To make the variation trends of $O_3$ clearer, we re-plotted Figure 2 with the linear [$O_3$].

According to the updated Figure 2, $O_3$ was identified to be increased from 2007 to 2013, which however decreased between 2013 and 2017, consistent with the previous results. The patterns were discernible from both the continuous measurements of $O_3$ and the $O_3$ observed in the three VOC sampling periods (Lines 323-348, pages 14-15 in the manuscript). It cannot be denied that the meteorological conditions played roles in influencing the $O_3$ trends. For example, the lower $O_3$ in the autumn of 2016 could be partially explained by the less frequent tropical cyclones and continental anticyclones, as discussed in lines 375-388, pages 16-17. However, the simulation of in situ photochemistry in previous studies (Xue et al., 2014a; Wang et al., 2017a; Lyu et al., 2017b) and this paper also indicated the effects of air quality policies on $O_3$ variations in Hong Kong. In addition, a recent study (Li et al., 2019) published in PNAS (Proceedings of the National Academy of Sciences of the United States of America) demonstrated the significant effects of artificial interventions on $O_3$ pollution in China with the filtration of meteorological impacts. The statistically unchanged $O_3$ residues (differences between observed $O_3$ and $O_3$ driven by meteorological conditions) during $2013 - 2017$ in most areas of PRD corroborated our finding that the contribution of regional transport to $O_3$ in Hong Kong stopped increasing since 2013. This reinforces our confidence in the conclusions of this study.

[Figure]

Figure 2. Long-term trends of the observed $O_3$ at TC from 2007 to 2017. Hourly $O_3$ values on the VOC sampling days in the autumns of 2007, 2013 and 2016 are marked in red. The hourly variation rates of $O_3$ are converted to yearly rates in periods of 2007 − 2013 and 2013 − 2017.

For details, please refer to Figure 2.

iii) Some further discussion of the locally produced and regionally transported ozone (p5 line 109) would be helpful. The wind speeds are generally quite low, so the distances the precursors and ozone itself are transported is not large. How significant is precursor transport? Are the sources greater in some directions for the longer-lived precursors? How significant is this? The assertion that the regionally transported emissions can be determined by difference as asserted on p5 needs some moderation-the chemistry isn't linear and the simulations aren't 100% accurate, as demonstrated by some apparently negative regional sources.

We highly appreciate the thoughtful comments and suggestion.

The winds were indeed low at the sampling site, which were likely weakened by the surrounding buildings. In fact, the wind speeds monitored at the Hong Kong International Airport which was ~3 km to the north of the site were much higher (>4 m/s, see Table S6 for details), indicating that air pollutants could be transported from PRD to the area where the site located. As shown in Figure S2, some $O_3$ precursors, including both the longer-lived species (*e.g.* CO, ethyne, ethane and propane) and shorter-lived species (*e.g.* toluene) exhibited higher concentrations under some wind sectors (such as the northwest winds), relative to the average concentrations under the light winds (wind speeds < 2m/s). Therefore, regional transport at least made some contributions to the $O_3$ precursors at the sampling site. Considering that the concentrations of air pollutants measured under light winds represented the local emissions, regional transport elevated the concentrations of CO, ethyne, ethane, propane and toluene by up to 11.3%, 48.0%, 42.5%, 53.5% and 138.5%, respectively.

The regionally-transported $O_3$ precursors made some contributions to the in-situ $O_3$ production. Therefore, the locally-produced $O_3$ were somewhat overestimated by assigning the simulated $O_3$ as the locally produced $O_3$. In addition, the nonlinear chemistry and model uncertainty raised by the reviewer also caused some uncertainties in the locally-produced and regionally-transported $O_3$ determined in this study, which are discussed in Text S1.

However, we may clarify that the negative values of regionally-transported $O_3$ were not necessary to be only caused by the uncertainties, but also indicative of the quick dispersion and dilution of $O_3$ by the south winds from the relatively clean South China Sea in most cases.

Discussions have been added in the revised manuscript.

With the aid of a photochemical box model, the locally-produced and regionally-transported O$_3$, as well as their variation trends, were determined (see section 2.5).

The PBM-MCM simulates the in-situ O$_3$ photochemistry based on the observed O$_3$ precursors. Figure S2 shows the average mixing ratios of some O$_3$ precursors in different wind sectors. The higher levels of CO, ethyne, ethane, propane and toluene under northwest winds indicated the transport of these species from PRD to Hong Kong. Meanwhile, O$_3$ might also be transported to Hong Kong. Text S1 discusses the determination of the locally produced and regionally transported O$_3$, as well as the uncertainties.

For details, please refer to lines 303-308, page 13.

Text S1. Determination of the locally produced and regionally transported O$_3$ and discussion on the uncertainties.

As an observation based model, PBM-MCM has been widely used to simulate the in-situ O$_3$ production (Lam et al., 2013; Ling et al., 2014; Lyu et al., 2017b; Wang et al., 2017a). Therefore, the O$_3$ simulated by PBM-MCM can be regarded as the locally produced O$_3$, and the differences between the observed and simulated O$_3$ were taken as the regionally transported O$_3$. However, it should be noted that the observed concentrations of O$_3$ precursors could be partially built up by regional transport. For example, under the northwest winds, the average mixing ratios of CO (693.9±25.5 ppbv), ethyne (2.15±0.22 ppbv), ethane (2.31±0.25 ppbv), propane (2.97±0.51 ppbv) and toluene (2.42±0.52 ppbv) were the highest among all the wind sectors, surpassing their average concentrations under light winds (wind speeds < 2 m/s) by 11.3%, 48.0%, 42.5%, 53.5% and 138.5%, respectively. Since the PBM-MCM was constrained by the observed concentrations of O$_3$ precursors, the share of regionally transported $O_3$ precursors in the observations made contributions to the simulated $O_3$, which in fact represented a kind of regional transport. Therefore, the locally produced $O_3$ was to some extent overestimated in this way. Conversely, the regionally transported $O_3$ was underestimated. However, it is difficult to accurately quantify the contributions of regional transport to $O_3$ precursors at the site. Moreover, due to the non-linear relationships between $O_3$ and its precursors, we did not quantitatively evaluate the overestimation of the locally produced $O_3$ and the underestimation of the regionally transported $O_3$.

For details, please refer to Figure S2 and Text S1.

iv) HONO was not monitored in 2016 and the 2011 values were used. Some justification, based on measurements elsewhere, is provided. Figure 7 shows the OH formation and loss rates and demonstrates the importance of HONO in the early morning, especially in 2016. The potential error in this rate analysis should be emphasized.

We are grateful for this excellent comment.

It is true that HONO is an important source of OH, particularly in the early morning when the nightlong secondarily formed HONO and the primarily emitted HONO in the morning rush hours are subject to photolysis in the presence of sunlight. However, it is pity that HONO was not measured in the three sampling campaigns. To our best knowledge, the HONO concentrations at the sampling site were only reported by Xu et al. (2015), based on the measurements in the autumn of 2011. Therefore, we used an average diurnal cycle of HONO measured by Xu et al. (2015) to constrain the HONO concentrations in the model. To evaluate the uncertainties of adopting the HONO concentrations in 2011, we calculated the average diurnal cycles of HONO in the 2007, 2013 and 2016 sampling campaigns (Figure S5), according to the diurnal patterns of HONO/$NO_x$ ratios determined at the same site (Xu et al., 2015) and real measurements of $NO_x$ concentrations. It was found that by adopting the values in 2011, the HONO concentrations were underestimated by 14.9% and 11.6% in 2007 and 2013, respectively, but were overestimated by 10.4% 2016. Further, this caused the maximum underestimation (overestimation) of the total OH production rate by 2.3±2.3% (21.6±5.2%) in 2007, 5.8±1.3% (3.4±1.0%) in 2013 and 5.7±1.3% (3.4±0.9%) in 2016. It should be noted that the maximum overestimation of the total OH production rate in 2007 (21.6±5.2%) occurred at 07:00 LT when the OH recycling was weak. During 08:00 – 19:00 LT, both the underestimation and overestimation of the simulated total OH production rates were less than 3%. Overall, by adopting the measured HONO in 2011 at TC, the simulations of OH formation and loss rates were not largely biased.

Besides, the HONO concentrations calculated from the HONO/$NO_x$ ratios and $NO_x$ concentrations also had certain uncertainties. Thus, we did not use the calculated HONO concentrations to constrain the model. In fact, the consistent input of HONO concentrations in the three sampling campaigns enabled us to look into the changes of $O_3$ and radical photochemistry induced by the other factors, such as VOCs, $NO_x$ and meteorological conditions.

Discussions are added in the main text and the supplement.

As stated in section 2.4.2, the average diurnal cycle of HONO measured at TC in 2011 was adopted in the simulations. To assess the uncertainties, we also calculated the HONO concentrations according to the measured HONO/$NO_x$ ratios and the $NO_x$ concentrations at TC

in the three sampling campaigns (Figure S5). The uncertainties in HONO concentrations and in the contributions of HONO to OH formation and loss rates are discussed in Text S3.

For details, please refer to lines 530-535, pages 24-25 and Text S3.

v)  This last point illustrates one of the deficiencies of the paper – the tendency to discuss the behavior of average values and reaction rates rather than delve into the diurnal variations, which show some interesting effects, some difficult to understand. For example, in Fig 7, the high $NO + HO_2$ rate shows that the chain length is quite long, especially in 2007 – the propagation rate considerably exceeds the initiation rates (mainly $O_3$ + alkene and $O^1D + H_2O$). The OH production from HONO is large balanced by HONO formation from OH + NO except in the early morning. On the other hand, the termination rate (OH + NO2) is quite large and significantly exceeds the initiation rates. Why is this? It needs some explanation. The effects are still there in the later years, although less pronounced.

Thank you for the insightful and professional comments.

First of all, we must admit a careless mistake we made in plotting Figure 7. The legends of the OH loss pathways were misplaced. Besides, the measured concentrations of formaldehyde were forgotten to input into the model, which are now used to constrain the model in the revised manuscript. The changes of the numbers in sections 3.2 and 3.3 are almost due to the re-simulations performed with the input of formaldehyde concentrations into the model. Please refer to the updated Figure 7 for the further review on the revised manuscript and the responses below.

The high reaction rates between $HO_2$ and NO at the sampling site were also reported in previous studies. For example, Xue et al. (2016) indicated that the formation rate of OH through $HO_2$ reacting with NO (12.5 ppbv/h) was nearly 5 times the total OH formation rate through the initial reactions (HONO photolysis, $O_3$ photolysis, $O_3$ + VOCs, $H_2O_2$ photolysis and $HNO_3$ photolysis) on an $O_3$ episode day at the same site. The high reaction rates between $HO_2$ and NO were mainly attributable to the sufficient $HO_2$ and NO at this suburban site. To verify, the formation and loss rates of OH at an island (WSI) more than 40 km away from Hong Kong (Wang et al., 2018a) were simulated. The simulation was performed on one day with low $NO_x$, when the island was dominated by the sea breezes. It was found that the reaction between $HO_2$ and NO only accounted for 42.7±0.2% of the OH formation rate, in contrast to 69.8±1.1% at TC.

The balance between HONO photolysis rates and the formation rates of HONO through NO reacting with OH was not either the truth, which was misleading information resulting from the mistakes in the Figure legend as we admitted. According to the updated Figure 7, the net OH production rates through HONO photolysis (the differences between the photolysis rates and formation rates of HONO) were 0.68±0.21 × $10^6$, 0.70±0.12 × $10^6$ and 0.87±0.12 × $10^6$ molecules $cm^{-3}$ $s^{-1}$ in the 2007, 2013 and 2016 sampling campaigns, respectively.

With regard to the unreasonably high reaction rates of the termination reaction (OH + $NO_2$), we apologize again, because this was also wrong information resulting from the misplaced Figure legend. As shown in the updated Figure 7, OH was primarily consumed by reacting with VOCs, and the termination reaction rates were at the same magnitude as the initial reaction rates. Specifically, the initial formation rates of OH were higher than the reaction rates of OH + $NO_2$ in the early morning (7:00 − 10:00 LT), which was reversed in the following hours of the day due to the enhanced OH productions through the propagation reaction. As a comparison, the model simulation indicated that the initial reaction rates were always higher than the termination reaction rates throughout a low $NO_x$ day on WSI, with the average rate of $3.3\pm0.02 \times 10^7$ and $0.2\pm0.002 \times 10^7$ molecules $cm^{-3} s^{-1}$, respectively. This was reasonable, because the reactions among peroxy radicals take over the role of $OH + NO_2$ in terminating the reactions in low $NO_x$ environments.

Based on the corrections and deeper analyses above, the discussions are also revised to delve into the diurnal variations rather than the average values of the OH formation and loss rates. Revisions have been made in the revised manuscript as follows.

Figure 7 presents the average diurnal profiles of the simulated OH and the formation and loss pathways dominating the recycling of OH during the three sampling periods, which roughly followed the typical pattern of the intensities of photochemical reactions, *i.e.* higher at noon and lower at the beginning and end of the day. On average, the simulated OH concentration was comparable ($p$=0.4) between the 2007 sampling campaign ($1.6\pm0.3\times10^6$ molecules $cm^{-3}$) and the 2013 sampling campaign ($1.5\pm0.2\times10^6$ molecules $cm^{-3}$), but it decreased ($p<0.05$) to $1.0\pm0.2\times10^6$ molecules $cm^{-3}$ in the 2016 sampling campaign.

As expected, the formation and loss rates of OH were basically balanced in all the cases. OH was mainly formed from the reaction of $HO_2+NO$, which accounted for $69.8\pm1.1\%$ of the total OH production rate over the three sampling campaigns. The photolysis of HONO ranked the second in supplying OH with the contribution of $22.0\pm1.4\%$. As stated in section 2.4.2, the average diurnal cycle of HONO measured at TC in 2011 was adopted in the simulations. To assess the uncertainties, we also calculated the HONO concentrations according to the measured HONO/NO$_x$ ratios and the NO$_x$ concentrations at TC in the three sampling campaigns (Figure S5). The uncertainties in HONO concentrations and in the contributions of HONO to OH formation and loss rates are discussed in Text S3. The formation of OH from HONO photolysis was most efficient in the early morning, which was explained by the morning peak of HONO concentration, due to the nocturnal heterogeneous formation and the vehicle emissions in morning rush hours. Apart from the two dominant pathways, O$_3$ photolysis (6.3±0.2%), ozonolysis of unsaturated VOCs (1.5±0.2%) and H$_2$O$_2$ photolysis (0.2±0.01%) also made some contributions to the formation of OH, with the highest rates at noon or in the early afternoon when the productions of O$_3$ and H$_2$O$_2$ were the most intensive. To sum up, the total formation rates of OH from the primary sources (photolysis of HONO, O$_3$ and H$_2$O$_2$, and ozonolysis of VOCs) were lower than the recycling rates of OH (HO$_2$+NO) throughout the day at TC, consistent with the results in Xue et al. (2016) simulated at the same site. The dominant role of HO$_2$+NO in OH formation at TC (average contribution of 69.8±1.1%) might be related to the abundant NO at this site. The same pathway was simulated and accounted for only 42.7±0.2% of the total OH formation rate at an island more than 40 km away from Hong Kong with very low NO concentrations, *i.e.* maximum of 0.56 ppbv (Wang et al., 2018a).

OH was mainly depleted by the reactions with VOCs (32.3±1.2%), NO$_2$ (31.9±0.9%), CO (19.3±0.6%) and NO (16.5±1.1%). The reaction rates of OH+NO (formation rates of HONO) had the highest values in the morning, approximately in line with the diurnal pattern of the HONO photolysis rates, which however were not completely balanced due to the constraint of HONO to observations in the model. The average net photolysis rates of HONO (differences between the HONO photolysis and formation rates) were $0.68\pm0.21\times10^6$, $0.70\pm0.12\times10^6$ and $0.87\pm0.12\times10^6$ molecules cm$^{-3}$ s$^{-1}$ in the 2007, 2013 and 2016 sampling campaigns, respectively. The losses of OH through the other pathways all exhibited the highest efficiencies at noon or in the early afternoon. It should be noted that the reaction between OH and $NO_2$ was not only the sink of OH but also a termination reaction in the photochemical system. In comparison, the termination reaction rates were lower than the OH formation rates from the primary sources (photolysis of HONO, $O_3$ and $H_2O_2$, and ozonolysis of VOCs) in the morning (7:00 – 10:00 LT), which were reversed in the following hours of the day due to the increases in OH concentrations.

For detail, please refer to lines 520-564, pages 24-26, Figure S5 and Text S3.

vi) It would help understand these arguments, and those given on page 25, if Fig S3 were shown, and discussed in more detail, in the main text. Why do the concentrations of $HO_2$ and $RO_2$ increase in the late afternoon in 2016?

Thank you for the suggestion.

We hope that most of the reviewer's concerns on these arguments have been addressed in the response to comment (v). The increases of $HO_2$ and $RO_2$ concentrations in the late afternoon were caused by the abnormally high concentrations of some VOCs or OVOCs on a few days. For example, on Nov. 17, 2007, the mixing ratios of toluene substantially increased from 0.38 ppbv at 15:00 LT to 10.4 ppbv at 16:00. The increases were also observed for the concentrations of xylenes, $n/i$-butanes, $n/i$-pentanes and some other VOCs. This led to the jumping-up of the simulated $RO_2$ concentration from $1.7\times10^7$ molecule cm$^{-3}$ s$^{-1}$ at 15:00 LT to

$4.5 \times 10^7$ molecule $cm^{-3}$ $s^{-1}$ at 16:00 LT. Similarly, the increases of VOC concentrations occurred on some days in 2016, such as ethylbenzene, xylenes and isoprene on Sept. 26, isoprene on Oct. 16 and propionaldehyde on Nov. 7.

Note that the NO concentrations at 16:00 LT were generally low, due to the photochemical consumption. As a result, $HO_2$ and $RO_2$ produced by the substantially elevated VOC concentrations could not be efficiently converted to OH and RO, causing the rebounding of these peroxy radicals. Since the sources of VOCs in the atmosphere were complicated and these high VOC concentrations could not be simply treated as outliers, we did nothing to the simulated $HO_2$ and $RO_2$ but explained the rebounding phenomenon in the caption of Figure S6. To avoid over complication of the main text, we did not move Figure S6 (original Figure S3) to the main text.

[Figure]

Figure S6. Average diurnal profiles of the simulated OH, $HO_2$ and $RO_2$ concentrations on VOC sampling days in 2007, 2013 and 2016. The rebounding of $HO_2$ and $RO_2$ concentrations in the late afternoon in 2007 and 2016 was caused by the substantial increases in the concentrations of some VOCs or OVOCs in several samples.

For details, please refer to Figure S6.

vii) The concentrations of oxygenates are very high (Table S4), yet their sources are not discussed in the main text. Are they produced by reaction or are there emission sources? Were the concentrations constrained, or were they simulated? Were high concentrations of other oxygenates generated in the simulations?

Thanks for the questions.

Yes, the concentrations of carbonyls (oxygenates) were on high levels. Though their sources were beyond the scope of this study, Guo et al. (2013b) and Ling et al. (2016b) indicated that secondary formation accounted for large fractions of the ambient formaldehyde (60-76%), acetaldehyde (45-53%) and acetone (~71%) in Hong Kong, leaving the rest to be contributed by 24-40%, 47-55% and ~29%, respectively. The concentrations of formaldehyde, acetaldehyde, acetone and propionaldehyde were constrained to observations in the model, while the other carbonyls were simulated by the model. The average simulated mixing ratios of methyl ethyl ketone (0.17±0.02 ppbv), methacrolein (0.12±0.01 ppbv), butyraldehyde (0.02±0.003 ppbv) and acrolein (0.02±0.004 ppbv) on all the VOC sampling days across the three sampling campaigns were in general lower than those reported in previous studies (Ho et al., 2002, 2007; Cheng et al., 2014), implying the sources of these OVOCs other than secondary formation. However, due to the low detection rates of these OVOCs in the samples, it is not reasonable to constrain their concentrations to the observations.

Clarifications have been made in the revised manuscript as follows.

Though previous studies (Guo et al., 2013b; Ling et al., 2016b) indicated that secondary formation dominated the sources of OVOCs in Hong Kong, the primary emissions could not be neglected. Therefore, formaldehyde, acetaldehyde, acetone and propionaldehyde with relatively high abundances were constrained to the observed concentrations in the model, while the other OVOCs with low concentrations and low detection rates were simulated by the model.

For details, please refer to lines 268-273, pages 12.

viii) Why were the BVOC concentrations so low in 2007 (Fig S4)? The temperature was slightly lower, but not by much, and the solar irradiance was high (Table S5).

Thanks for the question.

The low concentrations of BVOC in 2007 shown in Figure S7 (original Figure S4) were consistent with the low levels of isoprene (Table S5). Figure S8 shows the relationship between the common logarithm of isoprene mixing ratios and temperature. It is found that the higher temperatures corresponded to higher mixing ratios of isoprene, and the data points in 2007 did not deviate from the data points in 2013 and 2016, suggesting that the lower isoprene mixing ratios in 2007 were mainly attributable to the lower temperature.

In addition, the average wind speed in 2007 ($2.3 \pm 0.2$ m s$^{-1}$) was much higher ($p < 0.05$) than those in 2013 ($1.0 \pm 0.1$ m s$^{-1}$) and 2016 ($0.9 \pm 0.1$ m s$^{-1}$), with more frequent southeast winds in 2007 (62.8%). The strong winds from South China Sea might dilute isoprene emitted from the terrestrial plants, partially responsible for the low isoprene levels in 2007.

Explanation to the lower BVOCs in 2007 is given in the revised manuscript.

At last, the increase of BVOCs from 2007 to 2013 but comparable levels between 2013 and 2016 seemed to be related to the lower ($p < 0.05$) temperature in the 2007 sampling campaign (Figure S8 and Table S6). Besides, the more frequent (62.8%) southeast winds from SCS with higher wind speeds (2.3±0.2 m s$^{-1}$) might dilute BVOCs emitted from the terrestrial plants in the 2007 sampling campaign.

[Figure]

Figure S8. Relationship between the common logarithm of isoprene mixing ratios and temperature.

For details, please refer to lines 644-648, page 30 and Figure S8 and Table S6.

ix) While the analysis of the impact on O$_3$ of the emissions changes, via the scenario B simulations, is of considerable value, it is important, once again, to point out that the chemistry is non-linear, and the approach limits the quantitative, but not the qualitative, nature of the conclusions.

We thank for the good suggestion.

Indeed, due to the non-linearity of O$_3$ photochemistry, the contributions to the simulated O$_3$ of the individual sources determined by the subtraction approach were only the qualitative but not quantitative evaluations.

This has been clarified in the revised manuscript as follows.

It should be noted that due to the nonlinear relationships between $O_3$ and its precursors, the subtraction approach only qualitatively rather than quantitatively evaluated the contributions of VOC sources to $O_3$ production.

For details, please refer to lines 320-322, pages 14.

x)  Small point: the decrease in [OH] of less than 10% between 2007 and 2013 could hardly be described as significant (p24, line 502).

Accepted with thanks. It has been checked that the decrease of OH concentrations from 2007 to 2013 was insignificant, where $p=0.4$. Revisions have been made in the manuscript.

On average, the simulated OH concentration was comparable ($p=0.4$) between the 2007 sampling campaign ($1.6\pm0.3\times10^6$ molecules cm$^{-3}$) and the 2013 sampling campaign ($1.5\pm0.2\times10^6$ molecules cm$^{-3}$), but it decreased ($p<0.05$) to $1.0\pm0.2\times10^6$ molecules cm$^{-3}$ in the 2016 sampling campaign.

For details, please refer to line 524-527, page 24.

Some of these comments require relatively detailed responses, which may need to be consigned to the Supplement to avoid over complication of the main text. It is important, though, that the issues related to the detailed chemistry are resolved since much of the emphasis of the paper depends on the box model calculations.

Again, we express our sincere appreciation to the reviewer for his/her insightful and professional comments & suggestion. We realized the importance of addressing the problems related to the detailed chemistry in this paper and made efforts to improve the paper through correcting the mistakes (such as the big mistake in the legend of Figure 7), mining the data and deepening the discussion. As suggested, additional supporting information is provided in the Supplement to avoid the over complication of the main text. Hope that the revised manuscript is satisfactory to be published.

[revised manuscript text omitted]

As an observation based model, PBM-MCM has been widely used to simulate the in-situ O$_3$ production (Lam et al., 2013; Ling et al., 2014; Lyu et al., 2017b; Wang et al., 2017a). Therefore, the O$_3$ simulated by PBM-MCM can be regarded as the locally produced O$_3$, and the differences between the observed and simulated O$_3$ were taken as the regionally transported O$_3$. However, it should be noted that the observed concentrations of O$_3$ precursors could be partially built up by regional transport. For example, under the northwest winds, the average mixing ratios of CO (693.9$\pm$25.5 ppbv), ethyne (2.15$\pm$0.22 ppbv), ethane (2.31$\pm$0.25 ppbv), propane (2.97$\pm$0.51 ppbv) and toluene (2.42$\pm$0.52 ppbv) were the highest among all the wind sectors, surpassing their average concentrations under light winds (wind speeds < 2 m/s) by 11.3%, 48.0%, 42.5%, 53.5% and 138.5%, respectively. Since the PBM-MCM was constrained by the observed concentrations of O$_3$ precursors, the share of regionally transported O$_3$ precursors in the observations made contributions to the simulated O$_3$, which in fact represented a kind of regional transport. Therefore, the locally produced O$_3$ was to some extent overestimated in this way. Conversely, the regionally transported O$_3$ was underestimated. However, it is difficult to accurately quantify the contributions of regional transport to O$_3$ precursors at the site. Moreover, due to the non-linear relationships between

O$_3$ and its precursors, we did not quantitatively evaluate the overestimation of the locally produced O$_3$ and the underestimation of the regionally transported O$_3$.

Text S2. Set-up of the simulation scenarios

The base scenario (scenario A) was established to simulate the local production of O$_3$, with the observed concentrations of air pollutants (excluding O$_3$) as model inputs. The observed O$_3$ was not input because the simulated O$_3$ would be constrained to the observed values with the outputs exactly the same as the inputs otherwise. The scenario B was established to simulate O$_3$ under the assumption that a source of VOCs was totally removed. Namely, the VOCs emitted from a specific source were subtracted from the observed VOCs when allocating the model inputs. In this study, six sources of VOCs were identified (see section 3.4.1). Therefore, 6 sub-scenarios were included in scenario B, because the VOCs emitted from the individual sources were subtracted one by one. In this approach, the differences in simulated O$_3$ between scenario A and scenario B were the contributions of individual VOC sources to the local O$_3$ production.

Text S3. Uncertainties in HONO concentrations and the subsequent uncertainties in the contributions of HONO to OH formation and loss rates.

To evaluate the uncertainties of adopting the HONO concentrations in 2011 in the model, we calculated the average diurnal cycles of HONO in the 2007, 2013 and 2016 sampling campaigns (Figure S5), according to the diurnal patterns of HONO/NO$_x$ ratios determined at the same site (Xu et al., 2015) and the in-situ measurements of NO$_x$. It was found that by adopting the values in 2011, the HONO concentrations were underestimated by 14.9% and 11.6% in 2007 and 2013, respectively, but were overestimated by 10.4% 2016. Further, the sensitivity tests indicated that the maximum underestimation (overestimation) of the total OH

production rate was $2.3\pm2.3\%$ ($21.6\pm5.2\%$) in 2007, $5.8\pm1.3\%$ ($3.4\pm1.0\%$) in 2013 and

$5.7\pm1.3\%$ ($3.4\pm0.9\%$) in 2016. It should be noted that the maximum overestimation of the total OH production rate in 2007 ($21.6\pm5.2\%$) occurred at 07:00 LT when the OH recycling was weak. During 08:00 – 19:00 LT, both the underestimation and overestimation of the simulated total OH production rates were less than 3%. Therefore, it was concluded that the simulated OH formation and loss rates were not largely biased by adopting the measured

HONO at TC in 2011 in all the simulations.

Besides, the HONO concentrations calculated from the $HONO/NO_x$ ratios and $NO_x$

concentrations also had certain uncertainties. Thus, we did not use the calculated HONO

concentrations to constrain the model. In fact, the consistent input of the diurnal cycle of

HONO concentrations in the three sampling campaigns enabled us to look into the changes of

$O_3$ and radical photochemistry induced by the other factors, such as VOCs, $NO_x$ and meteorological conditions.

Table S1. Summary of the representative studies regarding $O_3$ pollution in Hong Kong.

| Reference | Site | Measurement period | Nature of monitoring | Target | Main conclusions |
|---|---|---|---|---|---|
| Cheng et al, 2010 | Wanqingsha (WQS), Guangdong and Tung Chung (TC), Hong Kong (HK) | Oct-Dec 2007 | Suburban | $O_3$ | $O_3$ formation was limited by VOCs at both sites. Carbonyls played important roles in photochemistry. |
| Cheng et al, 2013 | TC, HK | Sep 2007 and Sep 2008 | Suburban | $O_3$, VOCs | Major sources of VOCs in HK included consumer products, paint and printing solvent, road transport, and industrial, commercial, domestic and off-road transport. |
| Ding et al, 2004 | Pearl River Delta (PRD) | Sep 2001 | Large area | $O_3$ | $O_3$ pollution events in PRD were closely associated with sea-land breezes and tropical cyclones. |
| Guo et al, 2011 | WQS and TC | Oct-Dec 2007 | Suburban | VOCs | Solvent use, vehicular emissions, biomass burning, LPG usage and gasoline evaporation dominated the sources of VOCs in PRD. |
| Guo et al, 2013a | Tsuen Wan (TW) and Tai Mao Shan (TMS), HK | Sep-Nov 2010 | TW: Urban TMS: Mountainous | $O_3$ | Less NO titration, vertical transport, valley breeze and regional transport caused higher $O_3$ at the mountainous site. |
| Huang et al, 2005 | HK | 1999-2003 | Large area | $O_3$ | Tropical cyclones, continental anticyclones and troughs were conducive to $O_3$ pollution events in HK. |
| Lam et al, 2005 | TC, HK | Aug 1999 | Suburban | $O_3$, VOCs | Local thermal circulation under calm synoptic conditions trapped air pollutants, resulting in $O_3$ enhancement in HK. |
| Ling et al, 2013 | TC, HK | Sep-Nov 2010 | Suburban | $O_3$ | High $O_3$ in HK was a combined effect of both local formation and regional transport. |
| Ling et al, 2014 | TW and TMS, HK | Sep-Nov 2010 | TW: Urban TMS: Mountainous | $O_3$ | Different $O_3$ production and destruction pathways at two sites. More aged air masses at the mountainous site favored $O_3$ formation. |
| Lyu et al, 2016a | Multiple, HK | Sep 2013 and Sep 2014 | Urban Suburban Rural | VOCs | VOCs emitted from LPG-fueled vehicles significantly decreased at urban roadside sites. $O_3$ formation was limited by VOCs regardless of locations, while VOCs and $NO_x$ co-limited $O_3$ formation in rural areas. |
| Lyu et al, 2016b | Mong Kok (MK), HK | Jun 2011-May 2014 | Roadside | $O_3$, VOCs | Replacing catalytic converters in LPG-fueled vehicles led to substantial reductions of VOCs and $NO_x$ emissions. |
| Lyu et al, 2017a | TC, HK | 2005-2013 | Suburban | VOCs | VOCs emitted from solvent usage and diesel exhaust decreased in HK from 2005 to 2013. |

| Ou et al, 2015 | TC, HK | 2005-2013 | Suburban | VOCs | Vehicular exhaust, gasoline evaporation and LPG usage, consumer product and printing, architectural paints, and biogenic emissions were identified as the sources of VOCs in the study area. |
|---|---|---|---|---|---|
| So and Wang, 2003 | Multiple, HK | Jun 1999-May 2000 | Urban Suburban Rural | $O_3$ | Air quality in rural areas was frequently influenced by regional air masses, and was predominantly affected by local emissions in urban areas. |
| Wang et al, 2017a | TC, HK | 2005-2014 | Suburban | $O_3$ | Locally produced autumn $O_3$ decreased in HK, which was reversed by regionally-transported $O_3$ between 2005 and 2013. |
| Wang et al, 2018a | Wan Shan Island (WSI), GD | Aug-Nov 2013 | Rural | $O_3$ | $O_3$ formation switched from the $NO_x$-limited regime on low $O_3$ days to VOC-limited regime on high $O_3$ days over South China Sea. |
| Xu et al, 2008 | Linan, Zhejiang | Aug 1991-Jul 2006 | Rural | $O_3$ | Monthly highest 5% of $O_3$ increased from 1991 to 2006, with enhanced variability, likely due to the increased $NO_x$ emissions. |
| Xue et al, 2014a | TC, HK | Sep-Nov of 2002-2013 | Suburban | $O_3$ | Increase of regionally-transported $O_3$ offset the decrease of locally-produced $O_3$ and resulted in the increase of observed $O_3$ in the autumn in HK during 2002-2013. |
| Zhang et al, 2007 | Multiple, HK | Oct-Dec 2002 | Urban Suburban Rural | $O_3$ | 50-100% of $O_3$ enhancement during $O_3$ episodes in HK was explained by local photochemical formation, following the oxidation of anthropogenic VOCs. |

[revised manuscript text omitted]

*HKIA: Hong Kong International Airport.

Table S7. Number of $O_3$ episode days identified under the tropical cyclone, continental anticyclone and low pressure trough in the autumns of 2007, 2013 and 2016.

| Year | Total No. of Episode | Tropical cyclone | Continental anti-cyclone | Low Pressure trough |
|---|---|---|---|---|
| 2007 | 15* | 8 (4 typhoons) | 8 | 1 |
| 2013 | 16 | 11 (5 typhoons) | 5 | 0 |
| 2016 | 5 | 4 (3 typhoons) | 0 | 1 |

*Two $O_3$ episode days were under the combined influence of tropical cyclone and continental anticyclone.

[Figure]

Figure S1. Average diurnal cycles of $O_3$ in the 2007, 2013 and 2016 sampling campaigns. $\Delta O_3$-1: $O_3$ decrease in the early morning driven by NO titration; $\Delta O_3$-2: photochemically formed $O_3$ in the daytime (diurnal cycle of $O_3$ in 2007 as an example).

[Figure]

[Figure]

Figure S2. Average mixing ratios of some $O_3$ precursors in different wind sectors at TC in the three sampling campaigns. The orange dotted lines represent average concentrations under light winds (wind speed < 2 m/s).

Figure S3. Number of $O_3$ episode days and non-$O_3$ episode days in the autumns of 2007, 2013 and 2016.

[Figure]

Figure S4. Relationship between the hourly observed $O_3$ and the wind fields at TC in the three sampling campaigns.

[Figure]

Figure S5. Diurnal cycles of HONO mixing ratios measured at TC in the autumn of 2011 and those calculated from the measured HONO/NO$_x$ ratios and NO$_x$ mixing ratios at the same site in the three sampling campaigns.

[Figure]

Figure S6. Average diurnal profiles of the simulated OH, HO$_2$ and RO$_2$ concentrations on VOC sampling days in 2007, 2013 and 2016. The rebounding of HO$_2$ and RO$_2$ concentrations in the late afternoon in 2007 and 2016 was caused by the substantial increases in the concentrations of some VOCs or OVOCs in several samples.

[Figure]

Figure S7. Total mixing ratio of VOCs emitted from each individual source extracted from PMF in the 2007, 2013 and 2016 sampling campaigns. The solid lines represent the linear regressions of the VOC mixing ratios against the sequence number of the samples, with the slope being converted to yearly rates.

[Figure]

Figure S8. Relationship between the common logarithm of isoprene mixing ratios and temperature.